# Learning Hidden Markov Models from Non-sequence Data via Tensor Decomposition

**Tzu-Kuo Huang**
Machine Learning Department
Carnegie Mellon University
Pittsburgh, PA 15213
tzukuoh@cs.cmu.edu

**Jeff Schneider**
Robotics Institute
Carnegie Mellon University
Pittsburgh, PA 15213
schneide@cs.cmu.edu

## Abstract

Learning dynamic models from observed data has been a central issue in many scientific studies or engineering tasks. The usual setting is that data are collected sequentially from trajectories of some dynamical system operation. In quite a few modern scientific modeling tasks, however, it turns out that reliable sequential data are rather difficult to gather, whereas out-of-order snapshots are much easier to obtain. Examples include the modeling of galaxies, chronic diseases such as Alzheimer's, or certain biological processes.

Existing methods for learning dynamic model from non-sequence data are mostly based on Expectation-Maximization, which involves non-convex optimization and is thus hard to analyze. Inspired by recent advances in spectral learning methods, we propose to study this problem from a different perspective: moment matching and spectral decomposition. Under that framework, we identify reasonable assumptions on the generative process of non-sequence data, and propose learning algorithms based on the tensor decomposition method [2] to *provably* recover first-order Markov models and hidden Markov models. To the best of our knowledge, this is the first formal guarantee on learning from non-sequence data. Preliminary simulation results confirm our theoretical findings.

## 1 Introduction

Learning dynamic models from observed data has been a central issue in many fields of study, scientific or engineering tasks. The usual setting is that data are collected sequentially from trajectories of some dynamical system operation, and the goal is to recover parameters of the underlying dynamic model. Although many research and engineering efforts have been devoted to that setting, it turns out that in quite a few modern scientific modeling problems, another situation is more frequently encountered: observed data are out-of-order (or partially-ordered) snapshots rather than full sequential samples of the system operation. As pointed out in [7, 8], this situation may appear in the modeling of celestial objects such as galaxies or chronic diseases such as Alzheimer's, because observations are usually taken from different trajectories (galaxies or patients) at unknown, arbitrary times. Or it may also appear in the study of biological processes, such as cell metabolism under external stimuli, where most measurement techniques are destructive, making it very difficult to repetitively collect observations from the same individual living organisms as they change over time. However, it is much easier to take single snapshots of multiple organisms undergoing the same biological process in a fully asynchronous fashion, hence the lack of timing information. Rabbat et al. [9] noted that in certain network inference problems, the only available data are sets of nodes *co-occurring* in random walks on the network without the order in which they were visited, and the goal is to reconstruct the network structure from such co-occurrence data. This problem is essentially about learning a first-order Markov chain from data lacking sequence information.

As one can imagine, dynamic model learning in a non-sequential setting is much more difficult than in the sequential setting and has not been thoroughly studied. One issue is that the notion of non-sequence data is vague because there can be many different generative processes resulting in non-sequence data. Without any restrictions, one can easily find a case where no meaningful dynamic model can be learnt. It is therefore important to figure out what assumptions on the data and the model would lead to successful learning. However, existing methods for non-sequential settings, e.g., [9, 11, 6, 8], do not shed much light on this issue because they are mostly based on Expectation-Maximization (EM), which require non-convex optimization. Regardless of the assumptions we make, as long as the resulting optimization problem remains non-convex, formal analysis of learning guarantees is still formidable.

We thus propose to take a different approach, based on another long-standing estimation principle: *the method of moments* (MoM). The basic idea of MoM is to find model parameters such that the resulting moments match or resemble the empirical moments. For some estimation problems, this approach is able to give unique and consistent estimates while the maximum-likelihood method gets entangled in multiple and potentially undesirable local maxima. Taking advantage of this property, an emerging area of research in machine learning has recently developed MoM-based learning algorithms *with formal guarantees* for some widely used latent variable models, such as Gaussian mixture models[5], Hidden Markov models [3], Latent Dirichlet Allocation [1, 4], etc. Although many learning algorithms for these models exist, some having been very successful in practice, barely any formal learning guarantee was given until the MoM-based methods were proposed. Such breakthroughs seem surprising, but it turns out that they are mostly based on one crucial property: for quite a few latent variable models, the model parameters can be uniquely determined from *spectral decompositions* of certain low-order moments of observable quantities.

In this work we demonstrate that under the MoM and spectral learning framework, there are reasonable assumptions on the generative process of non-sequence data, under which *the tensor decomposition method* [2], a recent advancement in spectral learning, can provably recover the parameters of *first-order Markov models* and *hidden Markov models*. To the best of our knowledge, ours is the first work that provides formal guarantees for learning from non-sequence data. Interestingly, these assumptions bear much similarity to the usual idea behind *topic modeling*: with the bag-of-words representation which is *invariant to word orderings*, the task of inferring topics is almost impossible given *one single document* (no matter how long it is!), but becomes easier as more documents touching upon various topics become available. For learning dynamic models, what we need in the non-sequence data are *multiple sets* of observations, where each set contains independent samples generated from *its own initial distribution*, and the many different initial distributions together cover the entire (hidden) state space. In some of the aforementioned scientific applications, such as biological studies, this type of assumptions might be realized by running multiple experiments with different initial configurations or amounts of stimuli.

The main body of the paper consists of four sections. Section 2 briefly reviews the essentials of the tensor decomposition framework [2]; Section 3 details our assumptions on non-sequence data, tensor-decomposition based learning algorithms, and theoretical guarantees; Section 4 reports some simulation results confirming our theoretical findings, followed by conclusions in Section 5. Proofs of theoretical results are given in the appendices in the supplementary material.

## 2 Tensor Decomposition

We mainly follow the exposition in [2], starting with some preliminaries and notations. A real $p$-th order tensor $A$ is a member of the tensor product space $\bigotimes_{i=1}^p \mathbb{R}^{m_i}$ of $p$ Euclidean spaces. For a vector $\mathbf{x} \in \mathbb{R}^m$, we denote by $\mathbf{x}^{\otimes p} := \mathbf{x} \otimes \mathbf{x} \otimes \cdots \otimes \mathbf{x} \in \bigotimes_{i=1}^p \mathbb{R}^m$ its $p$-th tensor power. A convenient way to represent $A \in \bigotimes_{i=1}^p \mathbb{R}^m$ is through a $p$-way array of real numbers $[A_{i_1 i_2 \cdots i_p}]_{1 \leq i_1, i_2, \ldots, i_p \leq m}$, where $A_{i_1 i_2 \cdots i_p}$ denotes the $(i_1, i_2, \ldots, i_p)$-th coordinate of $A$ with respect to a canonical basis. With this representation, we can view $A$ as a multi-linear map that, given a set of $p$ matrices $\{X_i \in \mathbb{R}^{m \times m_i}\}_{i=1}^p$, produces another $p$-th order tensor $A(X_1, X_2, \cdots, X_p) \in \bigotimes_{i=1}^p \mathbb{R}^{m_i}$ with the following $p$-way array representation:

$$A(X_1, X_2, \cdots, X_p)_{i_1 i_2 \cdots i_p} := \sum_{1 \leq j_1, j_2, \ldots, j_p \leq m} A_{j_1 j_2 \cdots j_p} (X_1)_{j_1 i_1} (V_2)_{j_2 i_2} \cdots (X_p)_{j_p i_p}. \quad (1)$$

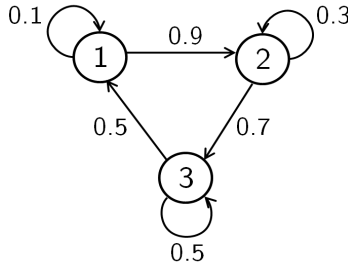

Figure 1: Running example of Markov chain with three states

In this work we consider tensors that are up to the third-order ($p \leq 3$) and, for most of the time, also *symmetric*, meaning that their $p$-way array representations are invariant under permutations of array indices. More specifically, we focus on second and third-order symmetric tensors in or slightly perturbed from the following form:

$$M_2 := \sum_{i=1}^{k} \omega_i \boldsymbol{\mu}_i \otimes \boldsymbol{\mu}_i, \quad M_3 := \sum_{i=1}^{k} \omega_i \boldsymbol{\mu}_i \otimes \boldsymbol{\mu}_i \otimes \boldsymbol{\mu}_i, \tag{2}$$

satisfying the following non-degeneracy conditions:

**Condition 1.** $\omega_i \geq 0 \ \forall \ 1 \leq i \leq k$, $\{\boldsymbol{\mu}_i \in \mathbb{R}^m\}_{i=1}^k$ *are linearly independent, and* $k \leq m$.

As described in later sections, the core of our learning task involves estimating $\{\omega_i\}_{i=1}^k$ and $\{\boldsymbol{\mu}_i\}_{i=1}^k$ from perturbed or noisy versions of $M_2$ and $M_3$. We solve this estimation problem with the tensor decomposition method recently proposed by Anandkumar et al. [2]. The algorithm and its theoretical guarantee are summarized in Appendix A. The key component of this method is a novel tensor power iteration procedure for factorizing a symmetric orthogonal tensor, which is robust against input perturbation.

# 3 Learning from Non-sequence Data

We first describe a generative process of non-sequence data for first-order Markov models and demonstrate how to apply tensor decomposition methods to perform consistent learning. Then we extend these ideas to hidden Markov models and provide theoretical guarantees on the sample complexity of the proposed learning algorithm. For notational conveniences we define the following vector-matrix cross product $\otimes_{d \in \{1,2,3\}}$ : $(\mathbf{v} \otimes_1 M)_{ijk} := v_i(M)_{jk}, (\mathbf{v} \otimes_2 M)_{ijk} = v_j(M)_{ik}, (\mathbf{v} \otimes_3 M)_{ijk} = v_k(M)_{ij}$. For a matrix $M$ we denote by $M_i$ its $i$-th column.

## 3.1 First-order Markov Models

Let $P \in [0,1]^{m \times m}$ be the transition probability matrix of a discrete, first-order, ergodic Markov chain with $m$ states and a unique stationary distribution $\boldsymbol{\pi}$. Let $P$ be of full rank and $\mathbf{1}^\top P = \mathbf{1}^\top$. To give a high-level idea of what makes it possible to learn $P$ from non-sequence data, we use the simple Markov chain with three states shown in Figure 1 as our running example, demonstrating step by step how to extend from a very restrictive generative setting of the data to a reasonably general setting, along with the assumptions made to allow consistent parameter estimation. In the usual setting where we have sequences of observations, say $\{\mathbf{x}^{(1)}, \mathbf{x}^{(2)}, \ldots\}$ with parenthesized superscripts denoting time, it is straightforward to consistently estimate $P$. We simply calculate the empirical frequency of consecutive pairs of states:

$$\widehat{P_{ij}} := \frac{\sum_t \ (\mathbf{x}^{(t+1)} = i, \mathbf{x}^{(t)} = j)}{\sum_t \ (\mathbf{x}^{(t)} = j)}.$$

Alternatively, suppose for each state $j$, we have an *i.i.d. sample* of its immediate next state $D_j := \{\mathbf{x}_1^{(1)}, \mathbf{x}_2^{(1)}, \ldots \mid \mathbf{x}^{(0)} = j\}$, where subscripts are data indices. Consistent estimation in this case is also easy: the empirical distribution of $D_j$ consistently estimates $P_j$, the $j$-th column of $P$. For

example, the Markov chain in Figure 1 may produce the following three samples, whose empirical distributions estimate the three columns of $P$ respectively:

$$
\begin{aligned}
D_1 &= \{2,1,2,2,2,2,2,2,2,2\} &\Rightarrow& \quad \widehat{P_1} = [0.1 \ \ 0.9 \ \ 0.0]^\top, \\
D_2 &= \{3,3,2,3,2,3,3,2,3,3\} &\Rightarrow& \quad \widehat{P_2} = [0.0 \ \ 0.3 \ \ 0.7]^\top, \\
D_3 &= \{1,1,3,1,3,3,1,3,3,1\} &\Rightarrow& \quad \widehat{P_3} = [0.5 \ \ 0.0 \ \ 0.5]^\top.
\end{aligned}
$$

A nice property of these estimates is that, unlike in the sequential setting, they do not depend on any particular ordering of the observations in each set. Nevertheless, such data are not quite non-sequenced because all observations are made at exactly the next time step. We thus consider the following generalization: for each state $j$, we have $D_j := \{\mathbf{x}_1^{(t_1)}, \mathbf{x}_2^{(t_2)}, \dots \mid \mathbf{x}^{(0)} = j\}$, i.e., independent samples of states drawn at *unknown* future times $\{t_1, t_2, \dots\}$. For example, our data in this setting might be

$$
\begin{aligned}
D_1 &= \{2,1,2,3,2,3,3,2,2,3\}, \\
D_2 &= \{3,3,2,3,2,1,3,2,3,1\}, \\
D_3 &= \{1,1,3,1,2,3,2,3,3,2\}.
\end{aligned}
\tag{3}
$$

Obviously it is hard to extract information about $P$ from such data. However, if we assume that the unknown times $\{t_i\}$ are i.i.d. random variables following some distribution independent of the initial state $j$, it can then be easily shown that $D_j$'s empirical distribution consistently estimates $T_j$, the $j$-th column of the the *expected transition probability matrix* $T := \mathbb{E}_t[P^t]$:

$$
\begin{aligned}
D_1 &= \{2,1,2,3,2,3,3,2,2,3\} &\Rightarrow& \quad \widehat{T_1} = [0.1 \ \ 0.5 \ \ 0.4]^\top, \\
D_2 &= \{3,3,2,3,2,1,3,2,3,1\} &\Rightarrow& \quad \widehat{T_2} = [0.2 \ \ 0.3 \ \ 0.5]^\top, \\
D_3 &= \{1,1,3,1,2,3,2,3,3,2\} &\Rightarrow& \quad \widehat{T_3} = [0.3 \ \ 0.3 \ \ 0.4]^\top.
\end{aligned}
$$

In general there exist many $P$'s that result in the same $T$. Therefore, as detailed later, we make a specific distributional assumption on $\{t_i\}$ to enable unique recovery of the transition matrix $P$ from $T$ (Assumption A.1). Next we consider a further generalization, where the unknowns are not only the time stamps of the observations, but also the initial state $j$. In other words, we only know each set was generated from the same initial state, but do not know the actual initial state. In this case, the empirical distributions of the sets consistently estimate the columns of $T$ in some *unknown permutation* $\Pi$:

$$
\begin{aligned}
D_{\Pi(3)} &= \{1,1,3,1,2,3,2,3,3,2\} &\Rightarrow& \quad \widehat{T_{\Pi(3)}} = [0.3 \ \ 0.3 \ \ 0.4]^\top. \\
D_{\Pi(2)} &= \{3,3,2,3,2,1,3,2,3,1\} &\Rightarrow& \quad \widehat{T_{\Pi(2)}} = [0.2 \ \ 0.3 \ \ 0.5]^\top, \\
D_{\Pi(1)} &= \{2,1,2,3,2,3,3,2,2,3\} &\Rightarrow& \quad \widehat{T_{\Pi(1)}} = [0.1 \ \ 0.5 \ \ 0.4]^\top.
\end{aligned}
$$

In order to be able to identify $\Pi$, we will again resort to randomness and assume the unknown initial states are random variables following a certain distribution (Assumption A.2) so that the data carry information about $\Pi$. Finally, we generalize from a single unknown initial state to an unknown *initial state distribution*, where each set of observations $D := \{\mathbf{x}_1^{(t_1)}, \mathbf{x}_2^{(t_2)}, \dots \mid \boldsymbol{\pi}^{(0)}\}$ consists of independent samples of states drawn at random times from some unknown initial state distribution $\boldsymbol{\pi}^{(0)}$. For example, the data may look like:

$$
\begin{aligned}
D_{\boldsymbol{\pi}_1^{(0)}} &= \{1,3,3,1,2,3,2,3,3,2\}, \\
D_{\boldsymbol{\pi}_2^{(0)}} &= \{3,1,2,3,2,1,3,2,3,1\}, \\
D_{\boldsymbol{\pi}_3^{(0)}} &= \{2,1,2,3,3,3,3,1,2,3\},
\end{aligned}
$$

$$\vdots$$

With this final generalization, most would agree that the generated data are non-sequenced and that the generative process is flexible enough to model the real-world situations described in Section 1. However, simple estimation with empirical distributions no longer works because each set may now contain observations from multiple initial states. This is where we take advantage of the tensor

decomposition framework outlined in Section 2, which requires proper assumptions on the initial state distribution $\boldsymbol{\pi}^{(0)}$ (Assumption A.3).

Now we are ready to give the definition of our entire generative process. Assume we have $N$ sets of non-sequence data each containing $n$ observations, and each set of observations $\{\mathbf{x}_i\}_{i=1}^n$ were independently generated by the following:

- Draw an initial distribution
  $$\boldsymbol{\pi}^{(0)} \sim \mathsf{Dirichlet}(\boldsymbol{\alpha}), \qquad\qquad\qquad\qquad\qquad \text{(Assumption A.3)}$$
  $$\mathbb{E}[\boldsymbol{\pi}^{(0)}] = \boldsymbol{\alpha}/(\textstyle\sum_{i=1}^m \alpha_i) = \boldsymbol{\pi}, \quad \pi_i \neq \pi_j \ \forall \ i \neq j. \qquad \text{(Assumption A.2)}$$
- For $i = 1, \ldots, n$,
  - Draw a discrete time $t_i \sim \mathsf{Geometric}(r), \ t_i \in \{1, 2, 3, \ldots\}$. $\quad$ (Assumption A.1)
  - Draw an initial state $\mathbf{s}_i \sim \mathsf{Multinomial}(\boldsymbol{\pi}_0), \ \mathbf{s}_i \in \{0, 1\}^m$.
  - Draw an observation $\mathbf{x}_i \sim \mathsf{Multinomial}(P^{t_i} \mathbf{s}_i), \ \mathbf{x}_i \in \{0, 1\}^m$.

The above generative process has several properties. First, all the data points in the same set share the same initial state distribution but can have different initial states; the initial state distribution varies across different sets and yet centers at the stationary distribution of the Markov chain. As mentioned in Section 1, this may be achieved in biological studies by running multiple experiments with different input stimuli, so the data collected in the same experiment can be assumed to have the same initial state distribution. Second, each data point is drawn from an independent trajectory of the Markov chain, a similar situation in the modeling of galaxies or Alzheimer's, and random time steps could be used to compensate for individual variations in speed: a small/large $t_i$ corresponds to a slowly/fast evolving individual object. Finally, the geometric distribution can be interpreted as an overall measure of the magnitude of speed variation: a large success probability $r$ would result in many small $t_i$', meaning that most objects evolve at similar speeds, while a small $r$ would lead to $t_i$'s taking a wide range of values, indicating a large speed variation.

To use the tensor decomposition method in Appendix A, we need the tensor structure (2) in certain low-order moments of observed quantities. The following theorem identifies such quantities:

**Theorem 1.** *Define the expected transition probability matrix* $T := \mathbb{E}_t[P^t] = rP(I - (1-r)P)^{-1}$ *and let* $\alpha_0 := \sum_i \alpha_i, C_2 := \mathbb{E}[\mathbf{x}_1 \mathbf{x}_2^\top]$ *and* $C_3 := \mathbb{E}[\mathbf{x}_1 \otimes \mathbf{x}_2 \otimes \mathbf{x}_3]$. *Then the following holds:*

$$\mathbb{E}[\mathbf{x}_1] = \boldsymbol{\pi}, \quad C_2 = \frac{1}{\alpha_0+1} T \mathsf{diag}(\boldsymbol{\pi}) T^\top + \frac{\alpha_0}{\alpha_0+1} \boldsymbol{\pi}\boldsymbol{\pi}^\top, \tag{4}$$

$$C_3 = \frac{2}{(\alpha_0+2)(\alpha_0+1)} \sum_i \pi_i T_i^{\otimes 3} + \frac{\alpha_0}{\alpha_0+2} \sum_{d=1}^3 \boldsymbol{\pi} \otimes_d C_2 - \frac{2\alpha_0^2}{(\alpha_0+2)(\alpha_0+1)} \boldsymbol{\pi}^{\otimes 3}, \tag{5}$$

$$M_2 := (\alpha_0 + 1)C_2 - \alpha_0 \boldsymbol{\pi}\boldsymbol{\pi}^\top = T\mathsf{diag}(\boldsymbol{\pi})T^\top, \tag{6}$$

$$M_3 := \frac{(\alpha_0+2)(\alpha_0+1)}{2} C_3 - \frac{(\alpha_0+1)\alpha_0}{2} \sum_{d=1}^3 \boldsymbol{\pi} \otimes_d C_2 + \alpha_0^2 \boldsymbol{\pi}^{\otimes 3} = \sum_i \pi_i T_i^{\otimes 3}. \tag{7}$$

The proof is in Appendix B.1, which relies on the special structure in the moments of the Dirichlet distribution (Assumption A.3). It is clear that $M_2$ and $M_3$ have the desired tensor structure. Assuming $\alpha_0$ is known, we can form estimates $\widehat{M_2}$ and $\widehat{M_3}$ by computing empirical moments from the data. Note that the $\mathbf{x}_i$'s are exchangeable, so we can use all pairs and triples of data points to compute the estimates. Interestingly, these low-order moments have a very similar structure to those in Latent Dirichlet Allocation [1]. Indeed, according to our generative process, we can view a set of non-sequence data points as a document generated by an LDA model with the expected transition matrix $T$ as the topic matrix, the stationary distribution $\boldsymbol{\pi}$ as the topic proportions, and most importantly, the states as *both the words and the topics*. The last property is what distinguishes our generative process from a general LDA model: because both the words and the topics correspond to the states, the topic matrix is no longer invariant to column permutations. Since the tensor decomposition method may return $\widehat{T}$ under any column permutation, we need to recover the correct matching between its rows and columns. Note that the $\widehat{\boldsymbol{\pi}}$ returned by the tensor decomposition method undergoes the same permutation as $\widehat{T}$'s columns. Because all $\pi_i$'s have different values by Assumption A.2, we may recover the correct matching by sorting both the returned $\widehat{\boldsymbol{\pi}}$ and the mean $\bar{\boldsymbol{\pi}}$ of all data.

A final issue is estimating $P$ and $r$ from $\widehat{T}$. This is in general difficult even when the exact $T$ is available because multiple choices of $P$ and $r$ may result in the same $T$. However, if the true transition matrix $P$ has at least one zero entry, then unique recovery is possible:

**Theorem 2.** *Let $P^*$, $r^*$, $T^*$ and $\boldsymbol{\pi}^*$ denote the true values of the transition probability matrix, the success probability, the expected transition matrix, and the stationary distribution, respectively. Assume that $P^*$ is ergodic and of full rank, and $P_{ij}^* = 0$ for some $i$ and $j$. Let $\mathcal{S} := \{\lambda/(\lambda-1) \mid \lambda$ is a real negative eigenvalue of $T^*\} \cup \{0\}$. Then the following holds:*

- $0 \leq \max(\mathcal{S}) < r^* \leq 1$.

- *For all $r \in (0,1] \setminus \mathcal{S}$, $P(r) := (rI + (1-r)T^*)^{-1}T^*$ is well-defined and*

$$\mathbf{1}^\top P(r) = \mathbf{1}^\top, \ P(r)\boldsymbol{\pi}^* = \boldsymbol{\pi}^*, \ P^* = P(r^*),$$
$$P(r)_{ij} \geq 0 \ \forall \, i,j \quad \Longleftrightarrow \quad r \geq r^*.$$

*That is, $P(r)$ is a stochastic matrix if and only if $r \geq r^*$.*

The proof is in Appendix C. This theorem indicates that we can determine $r^*$ from $T^*$ by doing bi-section on $(0,1]$. But this approach fails when we replace $T^*$ by an estimate $\widehat{T}$ because even $\widehat{P}(r^*)$ might contain negative values. A more practical estimation procedure is the following: for each value of $r$ in a decreasing sequence starting from 1, project $\widehat{P}(r) := (rI + (1-r)\widehat{T})^{-1}\widehat{T}$ onto the space of stochastic matrices and record the projection distance. Then search in the sequence of projection distances for the first sudden increase[1] starting from 1, and take the corresponding value of $r$ and projected $\widehat{P}(r)$ as our estimates.

Assuming the true $r$ and $\alpha_0$ are known, with the empirical moments being consistent estimators for the true moments and the tensor decomposition method guaranteed to return accurate estimates under small input perturbation, we can conclude that the estimates described above will converge (with high probability) to the true quantities as the sample size $N$ increases. We give sample complexity bound on estimation error in the next section for hidden Markov models.

## 3.2 Hidden Markov Models

Let $P$ and $\boldsymbol{\pi}$ now be defined over the hidden discrete state space of size $k$ and have the same properties as the first-order Markov model. The generative process here is almost identical to (and therefore share the same interpretation with) the one in Section 3.1, except for an extra mapping from the discrete hidden state to a continuous observation space:

- Draw a state indicator vector $\mathbf{h}_i \sim \mathsf{Multinomial}(P^{t_i}\mathbf{s}_i), \mathbf{h}_i \in \{0,1\}^k$.

- Draw an observation: $\mathbf{x}_i = U\mathbf{h}_i + \boldsymbol{\epsilon}_i$, where $U \in \mathbb{R}^{m \times k}$ denotes a rank-$k$ matrix of mean observation vectors for the $k$ hidden states, and the random noise vectors $\boldsymbol{\epsilon}_i$'s are i.i.d satisfying $\mathbb{E}[\boldsymbol{\epsilon}_i] = \mathbf{0}$ and $\mathrm{Var}[\boldsymbol{\epsilon}_i] = \sigma^2 I$.

Note that a spherical covariance[2] is required for the tensor decomposition method to be applicable. The low-order moments that lead to the desired tensor structure are given in the following:

**Theorem 3.** *Define the expected hidden state transition matrix $T := \mathbb{E}_t[P^t] = rP(I - (1-r)P)^{-1}$ and let $\alpha_0 := \sum_i \alpha_i, V_1 := \mathbb{E}[\mathbf{x}_1], V_2 := \mathbb{E}[\mathbf{x}_1\mathbf{x}_1^\top], V_3 := \mathbb{E}[\mathbf{x}_1^{\otimes 3}], C_2 := \mathbb{E}[\mathbf{x}_1\mathbf{x}_2^\top]$ and $C_3 := \mathbb{E}[\mathbf{x}_1 \otimes \mathbf{x}_2 \otimes \mathbf{x}_3]$. Then the following holds:*

$$V_1 = U\boldsymbol{\pi}, \quad V_2 = U\mathsf{diag}(\boldsymbol{\pi})U^\top + \sigma^2 I, \quad V_3 = \sum_i \pi_i U_i^{\otimes 3} + \sum_{d=1}^3 V_1 \otimes_d (\sigma^2 I),$$

$$M_2 := V_2 - \sigma^2 I = U\mathsf{diag}(\boldsymbol{\pi})U^\top, \quad M_3 := V_3 - \sum_{d=1}^3 V_1 \otimes_d (\sigma^2 I) = \sum_i \pi_i U_i^{\otimes 3},$$

$$C_2 = \frac{1}{\alpha_0+1}UT\mathsf{diag}(\boldsymbol{\pi})(UT)^\top + \frac{\alpha_0}{\alpha_0+1}V_1 V_1^\top,$$

$$C_3 = \frac{2}{(\alpha_0+2)(\alpha_0+1)}\sum_i \pi_i (UT)_i^{\otimes 3} + \frac{\alpha_0}{\alpha_0+2}\sum_{d=1}^3 V_1 \otimes_d C_2 - \frac{2\alpha_0^2}{(\alpha_0+2)(\alpha_0+1)}V_1^{\otimes 3}$$

$$M_2' := (\alpha_0+1)C_2 - \alpha_0 V_1 V_1^\top = UT\mathsf{diag}(\boldsymbol{\pi})(UT)^\top,$$

$$M_3' := \frac{(\alpha_0+2)(\alpha_0+1)}{2}C_3 - \frac{(\alpha_0+1)\alpha_0}{2}\sum_{d=1}^3 V_1 \otimes_d C_2 + \alpha_0^2 V_1^{\otimes 3} = \sum_i \pi_i (UT)_i^{\otimes 3}.$$

**Algorithm 1** Tensor decomposition method for learning HMM from non-sequence data

---

**input** $N$ sets of non-sequence data points, the success probability $r$, the Dirichlet parameter $\alpha_0$, the number of hidden states $k$, and numbers of iterations $\mathsf{L}$ and $\mathsf{N}$.

**output** Estimates $\widehat{\boldsymbol{\pi}}$, $\widetilde{P}$ and $\widetilde{U}$ possibly under permutation of state labels.

1: Compute empirical averages $\widehat{V_1}, \widehat{V_2}, \widehat{V_3}, \widehat{C_2}, \widehat{C_3}$, and $\widehat{\sigma^2} := \lambda_{\min}(\widehat{V_2} - \widehat{V_1}\widehat{V_1}^\top)$.
2: Compute $\widehat{M_2}, \widehat{M_3}, \widehat{M_2'}, \widehat{M_3'}$
3: Run Algorithm A.1 (Appendix A) on $\widehat{M_2}$ and $\widehat{M_3}$ with the number of hidden states $k$ to obtain a symmetric tensor $\widehat{\mathcal{T}} \in \mathbb{R}^{k \times k \times k}$ and a whitening transformation $\widehat{W} \in \mathbb{R}^{m \times k}$.
4: Run Algorithm A.2 (Appendix A) $k$ times each with numbers of iterations $\mathsf{L}$ and $\mathsf{N}$, the input tensor in the first run set to $\widehat{\mathcal{T}}$ and in each subsequent run set to the deflated tensor returned by the previous run, resulting in $k$ pairs of eigenvalue/eigenvector $\{(\hat{\lambda}_i, \widehat{\mathbf{v}}_i)\}_{i=1}^k$.
5: Repeat Steps 4 and 5 on $\widehat{M_2'}$ and $\widehat{M_3'}$ to obtain $\widehat{\mathcal{T}'}, \widehat{W'}$ and $\{(\hat{\lambda}_i', \widehat{\mathbf{v}}_i')\}_{i=1}^k$.
6: Match $\{(\hat{\lambda}_i, \widehat{\mathbf{v}}_i)\}_{i=1}^k$ with $\{(\hat{\lambda}_i', \widehat{\mathbf{v}}_i')\}_{i=1}^k$ by sorting $\{\hat{\lambda}_i\}_{i=1}^k$ and $\{\hat{\lambda}_i'\}_{i=1}^k$.
7: Obtain estimates of HMM parameters:

$$\widehat{UT} := (\widehat{W'})^\dagger \widehat{V'}\widehat{\Lambda'}, \quad \widehat{U} := (\widehat{W}^\top)^\dagger \widehat{V}\widehat{\Lambda},$$

$$\widehat{P} := (r\widehat{U} + (1-r)\widehat{UT})^\dagger \widehat{UT}, \quad \widehat{\boldsymbol{\pi}} := [\hat{\lambda}_1'^{-2} \cdots \hat{\lambda}_k'^{-2}]^\top,$$

where $\widehat{V} := [\widehat{\mathbf{v}}_1 \cdots \widehat{\mathbf{v}}_k], \widehat{\Lambda} := \mathsf{diag}([\hat{\lambda}_1 \cdots \hat{\lambda}_k]^\top)$; $\widehat{V'}$ and $\widehat{\Lambda'}$ are defined in the same way.
8: (Optional) Project $\widehat{\boldsymbol{\pi}}$ onto the simplex and $\widehat{P}$ onto the space of stochastic matrices.

---

The proof is in Appendix B.2. This theorem suggests that, unlike first-order Markov models, HMMs require *two* applications of the tensor decomposition methods, one on $M_2$ and $M_3$ for extracting the mean observation vectors $U$, and the other on $M_2'$ and $M_3'$ for extracting the matrix product $UT$. Another issue is that the estimates for $M_2$ and $M_3$ require an estimate for the noise variance $\sigma^2$, which is not directly observable. Nevertheless, since $M_2$ and $M_3$ are in the form of low-order moments of spherical Gaussian mixtures, we may use the existing result (Theorem 3.2, [2]) to obtain an estimate $\widehat{\sigma}^2 = \lambda_{\min}(\widehat{V_2} - \widehat{V_1}\widehat{V_1}^\top)$. The situation regarding permutations of the estimates is also different here. First note that $P = (rU + (1-r)UT)^\dagger UT$, which implies that permuting the columns of $U$ and the columns of $UT$ in the same manner has the effect of permuting both the rows and the columns of $P$, essentially re-labeling the hidden states. Hence we can only expect to recover $P$ up to some simultaneous row and column permutation. By the assumption that $\pi_i$'s are all different, we can sort the two estimates $\widehat{\boldsymbol{\pi}}$ and $\widehat{\boldsymbol{\pi}}'$ to match the columns of $\widehat{U}$ and $\widehat{UT}$, and obtain $\widehat{P}$ if $r$ is known. When $r$ is unknown, a similar heuristics to the one for first-order Markov models can be used to estimate $r$, based on the fact that $P = (rU + (1-r)UT)^\dagger UT = (rI + (1-r)T)^{-1}T$, suggesting that Theorem 2 remains true when expressing $P$ by $U$ and $UT$.

Algorithm 1 gives the complete procedure for learning HMM from non-sequence data. Combining the perturbation bounds of the tensor decomposition method (Appendix A), the whitening procedure (Appendix D.1) and the matrix pseudoinverse [10], and concentration bounds on empirical moments (Appendix D.3), we provide a sample complexity analysis:

**Theorem 4.** *Suppose the numbers of iterations $\mathsf{N}$ and $\mathsf{L}$ for Algorithm A.2 satisfy the conditions in Theorem A.1 (Appendix A), and the number of hidden states $k$, the success probability $r$, and the Dirichlet parameter $\alpha_0$ are all given. For any $\eta \in (0,1)$ and $\epsilon > 0$, if the number of sets $N$ satisfies*

$$N \geq \frac{12 \max(k^2, m)m^3\nu^3(\alpha_0 + 2)^2(\alpha_0 + 1)^2}{\eta}.$$

$$\max\left(\frac{225000}{\delta_{\min}^2}, \frac{4600}{\min(\sigma_k(M_2'), \sigma_k(M_2))^2}, \frac{42000c^2\sigma_1(UT)^2\max(\sigma_1(UT), \sigma_1(U), 1)^2}{\epsilon^2\sigma_k(rU + (1-r)UT)^4\min(\sigma_k(UT), \sigma_k(U), 1)^4}\right),$$

*where $c$ is some constant, $\nu := \max(\sigma^2 + \max_{i,j}(|U_{ik}|^2), 1), \delta_{\min} := \min_{i,j}|1/\sqrt{\pi_j} - 1/\sqrt{\pi_j}|,$ and $\sigma_i(\cdot)$ denotes the $i$-th largest singular value, then the $\widehat{P}$ and $\widehat{U}$ returned by Algorithm 1 satisfy*

$$Prob(\|P - \widehat{P}\| \leq \epsilon) \geq 1 - \eta \quad and \quad Prob\left(\|U - \widehat{U}\| \leq \frac{\epsilon\sigma_k(rU + (1-r)UT)^2}{6\sigma_1(UT)}\right) \geq 1 - \eta,$$

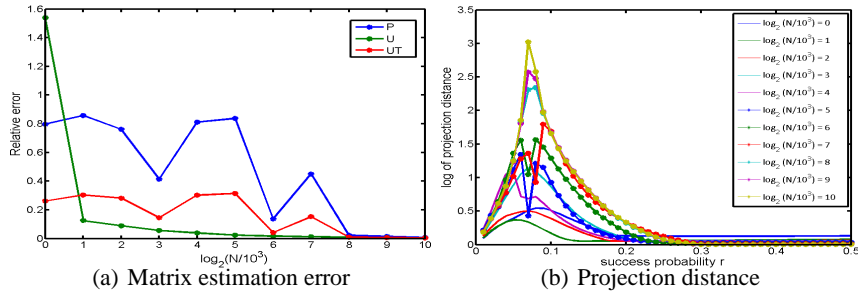

(a) Matrix estimation error        (b) Projection distance

Figure 2: Simulation results

*where $P$ and $U$ may undergo label permutation.*

The proof is in Appendix E. In this result, the sample size $N$ exhibits a fairly high-order polynomial dependency on $m, k, \epsilon^{-1}$ and scales with $1/\eta$ linearly instead of logarithmically, as is common in sample complexity results on spectral learning. This is because we do not impose any constraints on the observation model and simply use the Markov inequality for bounding the deviation in the empirical moments. If we make stronger assumptions such as boundedness or sub-Gaussianity, it is possible to use stronger, exponential tail bounds to obtain better sample complexity. Also worth noting is that $\delta_{\min}^{-2}$ acts as a threshold. As shown in our proof, as long as the operator norm of the tensor perturbation is sufficiently smaller than $\delta_{\min}$, which measures the gaps between different $\pi_i$'s, we can correctly match the two sets of estimated tensor eigenvalues. Lastly, the lower bound of $N$, as one would expect, depends on conditions of the matrices being estimated as reflected in the various ratios of singular values. An interesting quantity missing from the sample analysis is the size of each set $n$. To simplify the analysis we essentially assume $n = 3$, but understanding how $n$ might affect the sample complexity may have a critical impact in practice: when collecting more data, should we collect more sets or larger sets? What is the trade-off between them? This is an interesting direction for future work.

## 4 Simulation

Our HMM has $m = 40$ and $k = 5$ with Gaussian noise $\sigma^2 = 2$. The mean vectors $U$ were sampled from independent univariate standard normal and then normalized to lie on the unit sphere. The transition matrix $P$ contains one zero entry. For the generative process, we set $\alpha_0 = 1, r = 0.3, n = 1000$, and $N \in 1000\{2^0, 2^1, \ldots, 2^{10}\}$. The numbers of iterations for Algorithm A.2 were $\mathsf{N} = 200$ and $\mathsf{L} = 1000$. Figure 2(a) plots the relative matrix estimation error (in spectral norm) against the sample size $N$ for $P$, $U$, and $UT$ obtained by Algorithm 1 given the true $r$. It is clear that $U$ is the easiest to learn, followed by $UT$, and $P$ is the most difficult, and that all three errors converge to a very small value for sufficiently large $N$. Note that in Theorem 4 the bounds for $P$ and $U$ are different. With the model used here, the extra multiplicative factor in the bound for $U$ is less than 0.007, suggesting that $U$ is indeed easier to estimate than $P$. Figure 2(b) demonstrates the heuristics for determining $r$, showing projection distances (in logarithm) versus $r$. As $N$ increases, the take-off point gets closer to the true $r = 0.3$. The large peak indicates a pole (the set $S$ in Theorem 2).

## 5 Conclusions

We have demonstrated that under reasonable assumptions, tensor decomposition methods can provably learn first-order Markov models and hidden Markov models from non-sequence data. We believe this is the first formal guarantee on learning dynamic models in a non-sequential setting. There are several ways to extend our results. No matter what distribution generates the random time steps, tensor decomposition methods can always learn the expected transition probability matrix $T$. Depending on the application, it might be better to use some other distribution for the missing time. The proposed algorithm can be modified to learn discrete HMMs under a similar generative process. Finally, applying the proposed methods to real data should be the most interesting future direction.

## Footnotes

[1] Intuitively the jump should be easier to locate as $P$ gets sparser, but we do not have a formal result.

[2] We may allow different covariances $\sigma_j^2 I$ for different hidden states. See Section 3.2 of [2] for details.

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
