[Supplementary Material · sup.pdf]

# Supplemental Material for "Learning Hidden Markov Models from Non-sequence Data via Tensor Decomposition"

**Tzu-Kuo Huang**
Machine Learning Department
Carnegie Mellon University
Pittsburgh, PA 15213
tzukuoh@cs.cmu.edu

**Jeff Schneider**
Robotics Institute
Carnegie Mellon University
Pittsburgh, PA 15213
schneide@cs.cmu.edu

## A  Tensor Decomposition: Algorithm and Theoretical Guarantee

We start with the population version of the problem: suppose the noiseless $M_2$ and $M_3$ defined in (2) of the main text are available, and we want to recover $\omega_i{}_{i=1}^k$ and $\{\boldsymbol{\mu}_i\}_{i=1}^k$. First we perform a *whitening step* on them, as outlined in Algorithm A.1, to obtain a whitened, lower-dimensional tensor $\mathcal{T} \in \mathbb{R}^{k \times k \times k}$ and a whitening transformation $W \in \mathbb{R}^{m \times k}$ such that

$$\mathcal{T} \; := \; M_3(W, W, W) \; = \; \sum_{i=1}^k \omega_i (W^\top \boldsymbol{\mu}_i)^{\otimes 3} = \sum_{i=1}^k \frac{1}{\sqrt{\omega_i}} \widetilde{\boldsymbol{\mu}}_i^{\otimes 3},$$

where the vectors $\widetilde{\boldsymbol{\mu}}_i := \sqrt{\omega_i} W^\top \boldsymbol{\mu}_i$ form an orthonormal basis of $\mathbb{R}^k$ because $I = W^\top M_2 W = \sum_{i=1}^k W^\top (\sqrt{\omega_i} \boldsymbol{\mu}_i)(\sqrt{\omega_i} \boldsymbol{\mu}_i)^\top W = \sum_{i=1}^k \widetilde{\boldsymbol{\mu}}_i \widetilde{\boldsymbol{\mu}}_i^\top$. Hence, the symmetric tensor $\mathcal{T}$ has a so-called *orthogonal decomposition*, which may not exist for general symmetric tensors. Then by Theorem 4.3 of [2], which establishes the following results under Condition 1:

1. the set of *robust eigenvectors*[1] of $\mathcal{T}$ correspond exactly to $\{\widetilde{\boldsymbol{\mu}}_i\}_{i=1}^k$;
2. the eigenvalue associated with $\widetilde{\boldsymbol{\mu}}_i$ is equal to $1/\sqrt{\omega_i}$, $\forall\, 1 \le i \le k$;
3. if $(\mathbf{v}, \lambda)$ is a pair of robust eigenvector/eigenvalue of $\mathcal{T}$, then $\boldsymbol{\mu}_i = \lambda (W^\top)^\dagger \mathbf{v}$ for some $1 \le i \le k$, where $\dagger$ denotes the Moore-Penrose pseudo inverse;

we can reduce the original problem of recovering the structure in (2) into a robust tensor eigendecomposition problem. Motivated by the power iteration for matrix eigen computation, Anandkumar et al. [2] verify that starting from almost every vector $\boldsymbol{\theta}_0 \in \mathbb{R}^k$, the tensor power iteration $\boldsymbol{\theta}_t := \frac{\mathcal{T}(I, \boldsymbol{\theta}_{t-1}, \boldsymbol{\theta}_{t-1})}{\|\mathcal{T}(I, \boldsymbol{\theta}_{t-1}, \boldsymbol{\theta}_{t-1})\|}$, where $\|\cdot\|$ denotes the vector 2-norm, converges to some robust eigenvector of $\mathcal{T}$ at a quadratic rate, and therefore $k$ successive applications of the tensor power iteration with deflation result in all pairs of robust eigenvectors/eigenvalues.

In practice we almost never have the exact $M_2$ and $M_3$, but only noisy or perturbed versions $\widehat{M_2}$ and $\widehat{M_3}$, which are usually estimates from the data. Perturbation may destroy the nice tensor structure, so the reduced tensor $\widehat{\mathcal{T}}$ resulting from applying Algorithm A.1 to $\widehat{M_2}$ and $\widehat{M_3}$ may no longer be orthogonally decomposable, hindering the subsequent robust tensor eigendecomposition. Nevertheless, Anandkumar et al. [2] demonstrate that if the perturbation $E := \widehat{\mathcal{T}} - \mathcal{T}$ is a symmetric tensor with a small operator norm defined as $\|E\| := \sup_{\|\boldsymbol{\theta}\|=1} |E(\boldsymbol{\theta}, \boldsymbol{\theta}, \boldsymbol{\theta})|$, then $k$ successive applications of some *randomized* tensor power iteration coupled with deflation yield accurate estimates of all robust eigenvector/eigenvalue pairs with high probability. More precisely, they propose *the*

**Algorithm A.1** Whitening transformation

---

**input** A symmetric matrix $M_2 \in \mathbb{R}^{m \times m}$, a symmetric third-order tensor $M_3 \in \mathbb{R}^{m \times m \times m}$, and the target number of dimensions $k$.

**output** A reduced third-order tensor $\mathcal{T} \in \mathbb{R}^{k \times k \times k}$ and a whitening transformation $W \in \mathbb{R}^{m \times k}$.

1: Compute $W := QD^{-1/2}$, where $Q \in \mathbb{R}^{m \times k}$ denotes the top-$k$ orthonormal eigenvectors of $M_2$, and $D \in \mathbb{R}^{k \times k}$ is a diagonal matrix of the corresponding $k$ positive eigenvalues.
2: Compute $\mathcal{T} := M_3(W, W, W)$.

---

---

**Algorithm A.2** Robust tensor power method

---

**input** A symmetric tensor $\mathcal{T} \in \mathbb{R}^{k \times k \times k}$, number of iterations $\mathtt{L}, \mathtt{N}$.

**output** the estimated eigenvector/eigenvalue pair; the deflated tensor.

1: **for** $\tau = 1$ **to** $\mathtt{L}$ **do**
2:     Draw $\boldsymbol{\theta}_0^{(\tau)}$ uniformly at random from the unit sphere in $\mathbb{R}^k$.
3:     **for** $t = 1$ **to** $\mathtt{N}$ **do**
4: $$\boldsymbol{\theta}_t^{\tau} := \frac{\mathcal{T}(I, \boldsymbol{\theta}_{t-1}^{(\tau)}, \boldsymbol{\theta}_{t-1}^{(\tau)})}{\|\mathcal{T}(I, \boldsymbol{\theta}_{t-1}^{(\tau)}, \boldsymbol{\theta}_{t-1}^{(\tau)})\|}.$$
5:     **end for**
6: **end for**
7: Let $\tau^* := \arg\max_{1 \leq \tau \leq \mathtt{L}} \mathcal{T}(\boldsymbol{\theta}_{\mathtt{N}}^{(\tau)}, \boldsymbol{\theta}_{\mathtt{N}}^{(\tau)}, \boldsymbol{\theta}_{\mathtt{N}}^{(\tau)})$.
8: Do $\mathtt{N}$ power iteration updates (Line 4) starting from $\boldsymbol{\theta}_{\mathtt{N}}^{(\tau^*)}$ to obtain $\widehat{\boldsymbol{\theta}}$, and set $\hat{\lambda} := \mathcal{T}(\widehat{\boldsymbol{\theta}}, \widehat{\boldsymbol{\theta}}, \widehat{\boldsymbol{\theta}})$
9: **return** the estimated eigenvector/eigenvalue pair $(\widehat{\boldsymbol{\theta}}, \hat{\lambda})$; the deflated tensor $\mathcal{T} - \hat{\lambda}\widehat{\boldsymbol{\theta}}^{\otimes 3}$.

---

*Robust tensor power method* outlined in Algorithm A.2, which employs multiple random restarts, and provide a theoretical guarantee on its robustness against the input perturbation:

**Theorem A.1.** *(Theorem 5.1 of [2]) Let $\widehat{\mathcal{T}} = \mathcal{T} + E \in \mathbb{R}^{k \times k \times k}$, where $\mathcal{T}$ is a symmetric tensor with orthogonal decomposition $\mathcal{T} = \sum_{i=1}^{k} \lambda_i \mathbf{v}_i^{\otimes 3}$ where each $\lambda_i > 0$, $\{\mathbf{v}_1, \mathbf{v}_2, \ldots, \mathbf{v}_k\}$ is an orthonormal basis, and $E$ has operator norm $\epsilon := \|E\|$. Define $\lambda_{\min} := \min(\{\lambda_i\}_{i=1}^{k})$ and $\lambda_{\max} := \max(\{\lambda_i\}_{i=1}^{k})$. There exists universal constants $c_1, c_2, c_3 > 0$ such that the following holds. Pick any $\eta \in (0, 1)$, and suppose*

$$\epsilon \leq c_1 \cdot \frac{\lambda_{\min}}{k}, \qquad \mathtt{N} \geq c_2 \cdot \left(\log(k) + \log\log(\lambda_{\max}/\epsilon)\right), \qquad and$$

$$\sqrt{\frac{\ln(\mathtt{L}/\log_2(\frac{k}{\eta}))}{\ln(k)}} \cdot \left(1 - \frac{\ln(\ln(\mathtt{L}/\log_2(\frac{k}{\eta}))) + c_3}{4\ln(\mathtt{L}/\log_2(\frac{k}{\eta}))} - \sqrt{\frac{\ln(8)}{\ln(\mathtt{L}/\log_2(\frac{k}{\eta}))}}\right) \geq 1.02\left(1 + \sqrt{\frac{\ln(4)}{\ln(k)}}\right).$$

*(Note that the condition on $\mathtt{L}$ holds with $\mathtt{L} = poly(k)\log(1/\eta)$.) Suppose that Algorithm A.2 is iteratively called $k$ times with numbers of iterations $\mathtt{L}$ and $\mathtt{N}$, where the input tensor is $\widehat{\mathcal{T}}$ in the first call, and in each subsequent call, the input tensor is the deflated tensor returned by the previous call. Let $(\widehat{\mathbf{v}}_1, \hat{\lambda}_1), (\widehat{\mathbf{v}}_2, \hat{\lambda}_2), \ldots, (\widehat{\mathbf{v}}_k, \hat{\lambda}_k)$ be the sequence of estimated eigenvector/eigenvalue pairs returned in these $k$ calls. With probability at least $1 - \eta$, there exists a permutation $\rho$ on $\{1, \ldots, k\}$ such that*

$$\|\mathbf{v}_{\rho(j)} - \widehat{\mathbf{v}}_j\| \leq 8\epsilon/\lambda_{\rho(j)}, \quad |\lambda_{\rho(j)} - \hat{\lambda}_j| \leq 5\epsilon, \ \forall 1 \leq j \leq k, \quad and \quad \left\|\mathcal{T} - \sum_{j=1}^{k} \hat{\lambda}_j \widehat{\mathbf{v}}_j^{\otimes 3}\right\| \leq 55\epsilon.$$

This result, together with existing perturbation theory regarding the whitening procedure (e.g., Appendix C.1 of [1]), allow us to translate the perturbations in $\widehat{M_2}$ and $\widehat{M_3}$ into the estimation errors in $\omega_i$'s and $\boldsymbol{\mu}_i$'s, guaranteeing accurate estimation under small input perturbation.

# B Tensor Structure in Low-order Moments

Here we give proofs of theorems regarding tensor structures in low-order moments.

## B.1 Proof of Theorem 1

$$\mathbb{E}[\mathbf{x}_1] = \mathbb{E}_{\boldsymbol{\pi}_0}\mathbb{E}[\mathbf{x}_1 \mid \boldsymbol{\pi}_0] = \mathbb{E}_{\boldsymbol{\pi}_0}\mathbb{E}[P^{t_1}\mathbf{s}_1 \mid \boldsymbol{\pi}_0] = \mathbb{E}_{\boldsymbol{\pi}_0}[\mathbb{E}[P^{t_1}]\boldsymbol{\pi}_0] = T\boldsymbol{\pi} = \boldsymbol{\pi},$$

$$C_2 := \mathbb{E}[\mathbf{x}_1\mathbf{x}_2^\top] = \mathbb{E}_{\boldsymbol{\pi}_0}\mathbb{E}[P^{t_1}\mathbf{s}_1\mathbf{s}_2^\top(P^{t_2})^\top \mid \boldsymbol{\pi}_0] = \mathbb{E}_{\boldsymbol{\pi}_0}[\mathbb{E}[P^{t_1}]\mathbb{E}[\mathbf{s}_1\mathbf{s}_2^\top \mid \boldsymbol{\pi}_0]\mathbb{E}[(P^{t_2})^\top]]$$

$$= T\mathbb{E}_{\boldsymbol{\pi}_0}[\boldsymbol{\pi}_0\boldsymbol{\pi}_0^\top]T^\top = T\left(\frac{\mathsf{diag}(\boldsymbol{\pi})}{\alpha_0+1} + \frac{\alpha_0\boldsymbol{\pi}\boldsymbol{\pi}^\top}{\alpha_0+1}\right)T^\top = \frac{T\mathsf{diag}(\boldsymbol{\pi})T^\top}{\alpha_0+1} + \frac{\alpha_0\boldsymbol{\pi}\boldsymbol{\pi}^\top}{\alpha_0+1}, \quad (1)$$

$$C_3 := \mathbb{E}[\mathbf{x}_1 \otimes \mathbf{x}_2 \otimes \mathbf{x}_3] = \mathbb{E}_{\boldsymbol{\pi}_0}\mathbb{E}[(P^{t_1}\mathbf{s}_1) \otimes (P^{t_2}\mathbf{s}_2) \otimes (P^{t_3}\mathbf{s}_3) \mid \boldsymbol{\pi}_0]$$

$$= \mathbb{E}_{\boldsymbol{\pi}_0}[(T\boldsymbol{\pi}_0) \otimes (T\boldsymbol{\pi}_0) \otimes (T\boldsymbol{\pi}_0)] = \frac{\sum_i 2\pi_i T_i \otimes T_i \otimes T_i}{(\alpha_0+2)(\alpha_0+1)} + \frac{\alpha_0^2\boldsymbol{\pi} \otimes \boldsymbol{\pi} \otimes \boldsymbol{\pi}}{(\alpha_0+2)(\alpha_0+1)} \quad (2)$$

$$+ \frac{\alpha_0\left(\sum_{ij}\left(T_i \otimes T_i \otimes T_j + T_i \otimes T_j \otimes T_i + T_j \otimes T_i \otimes T_i\right)\pi_i\pi_j\right)}{(\alpha_0+2)(\alpha_0+1)}$$

$$= \frac{\sum_i 2\pi_i T_i \otimes T_i \otimes T_i}{(\alpha_0+2)(\alpha_0+1)} + \frac{\alpha_0(\boldsymbol{\pi} \otimes_3 C_2 + \boldsymbol{\pi} \otimes_2 C_2 + \boldsymbol{\pi} \otimes_1 C_2)}{\alpha_0+2} - \frac{2\alpha_0^2\boldsymbol{\pi} \otimes \boldsymbol{\pi} \otimes \boldsymbol{\pi}}{(\alpha_0+2)(\alpha_0+1)}, \quad (3)$$

The second equality in (1) and the two equalities (2) and (3) can be established by using the results on Dirichlet moments in Appendix B.1 of [1].

## B.2 Proof of Theorem 3

$$V_1 := \mathbb{E}[\mathbf{x}_1] = \mathbb{E}[U\mathbf{h}_1 + \epsilon_1] = U\mathbb{E}[P^{t_1}\mathbf{s}_1] = UT\mathbb{E}[\boldsymbol{\pi}_0] = U\boldsymbol{\pi}.$$

$$V_2 := \mathbb{E}[\mathbf{x}_1\mathbf{x}_1^\top] = \mathbb{E}[(U\mathbf{h}_1 + \epsilon_1)(U\mathbf{h}_1 + \epsilon_1)^\top]$$

$$= \mathbb{E}[U\mathbf{h}_1\mathbf{h}_1^\top U^\top] + \sigma^2 I = U\mathbb{E}[\mathsf{diag}(\mathbf{h}_1)]U^\top + \sigma^2 I$$

$$= U\mathbb{E}[\mathsf{diag}(P^{t_1}\mathbf{s}_1)]U^\top + \sigma^2 I = U\mathbb{E}[\mathsf{diag}(T\boldsymbol{\pi}_0)]U^\top + \sigma^2 I$$

$$= U\mathsf{diag}(\boldsymbol{\pi})U^\top + \sigma^2 I.$$

$$V_3 := \mathbb{E}[\mathbf{x}_1 \otimes \mathbf{x}_1 \otimes \mathbf{x}_1] = \mathbb{E}[(U\mathbf{h}_1 + \epsilon_1) \otimes (U\mathbf{h}_1 + \epsilon_1) \otimes (U\mathbf{h}_1 + \epsilon_1)]$$

$$= \mathbb{E}[(U\mathbf{h}_1) \otimes (U\mathbf{h}_1) \otimes (U\mathbf{h}_1)] + \mathbb{E}[(U\mathbf{h}_1) \otimes \epsilon_1 \otimes \epsilon_1] + \mathbb{E}[\epsilon_1 \otimes (U\mathbf{h}_1) \otimes \epsilon_1] + \mathbb{E}[\epsilon_1 \otimes \epsilon_1 \otimes (U\mathbf{h}_1)]$$

$$= \sum_i \pi_i U_i \otimes U_i \otimes U_i + V_1 \otimes_1 (\sigma^2 I) + V_1 \otimes_2 (\sigma^2 I) + V_1 \otimes_3 (\sigma^2 I).$$

$$C_2 := \mathbb{E}[\mathbf{x}_1\mathbf{x}_2^\top] = \mathbb{E}[(U\mathbf{h}_1 + \epsilon_1)(U\mathbf{h}_2 + \epsilon_2)^\top] = \mathbb{E}[U\mathbf{h}_1\mathbf{h}_2^\top U^\top]$$

$$= U\mathbb{E}[P^{t_1}\mathbf{s}_1\mathbf{s}_2^\top(P^{t_2})^T]U^\top = UT\mathbb{E}[\boldsymbol{\pi}_0\boldsymbol{\pi}_0^\top]T^\top U^\top = \frac{UT\mathsf{diag}(\boldsymbol{\pi})(UT)^\top}{\alpha_0+1} + \frac{\alpha_0 V_1 V_1^\top}{\alpha_0+1}. \quad (4)$$

$$C_3 := \mathbb{E}[\mathbf{x}_1 \otimes \mathbf{x}_2 \otimes \mathbf{x}_3] = \mathbb{E}[(U\mathbf{h}_1 + \epsilon_1) \otimes (U\mathbf{h}_2 + \epsilon_2) \otimes (U\mathbf{h}_3 + \epsilon_3)] = \mathbb{E}[(U\mathbf{h}_1) \otimes (U\mathbf{h}_2) \otimes (U\mathbf{h}_3)]$$

$$= \mathbb{E}[(UP^{t_1}\mathbf{s}_1) \otimes (UP^{t_2}\mathbf{s}_2) \otimes (UP^{t_3}\mathbf{s}_3)] = \mathbb{E}[(UT\boldsymbol{\pi}_0) \otimes (UT\boldsymbol{\pi}_0) \otimes (UT\boldsymbol{\pi}_0)]$$

$$= \frac{\sum_i 2\pi_i (UT)_i \otimes (UT)_i \otimes (UT)_i}{(\alpha_0+2)(\alpha_0+1)} + \frac{\alpha_0(V_1 \otimes_3 C_2 + V_1 \otimes_2 C_2 + V_1 \otimes_1 C_2)}{\alpha_0+2} - \frac{2\alpha_0^2 V_1 \otimes V_1 \otimes V_1}{(\alpha_0+2)(\alpha_0+1)}, \quad (5)$$

Again, the last equality in (4) and (5) can be established by using the results on Dirichlet moments in Appendix B.1 of [1].

# C  Proof of Theorem 2

We first prove the following lemma:

**Lemma 1.** *If $P(r) := (rI + (1-r)T^*)^{-1}T^*$ exists and is a stochastic matrix for some $r \in (0,1]$, then $P(r')$ exists and is a stochastic matrix for all $r' \in [r,1]$.*

*Proof.* Since $P(r)$ exists we can write $T^* = rP(r)(I - (1-r)P(r))^{-1}$. By assumption $P^*$ is invertible, so $T^*$ is invertible. We then have

$$r'(T^*)^{-1} + (1 - r')I = \frac{r'}{r}(P(r)^{-1} - (1-r)I) + (1 - r')I = \frac{r'}{r}P(r)^{-1}(I - (1 - r/r')P(r)),$$

which is invertible for all $r' \in [r, 1]$. Therefore, we can write

$$P(r') = (r'(T^*)^{-1} + (1 - r')I)^{-1} = \frac{r}{r'}P(r)(I - (1 - r/r')P(r))^{-1} = \mathbb{E}_t[P(r)],$$

where $t \sim \mathsf{Geometric}(r/r')$, showing that $P(r')$ is a stochastic matrix. $\qquad\square$

To prove Theorem 2 we begin by noting that $\mathcal{S}$ contains all values of $r$ for which $rI + (1 - r)T^*$ is singular. Therefore, $P(r)$ is well-defined and invertible for $r \in (0, 1] \setminus \mathcal{S}$. From the identity $T^*\boldsymbol{\pi}^* = \boldsymbol{\pi}^* = (rI + (1 - r)T^*)\boldsymbol{\pi}^*$ we have $P(r)\boldsymbol{\pi}^* = \boldsymbol{\pi}^*$, $r \notin \mathcal{S}$. Similarly, the identity $\mathbf{1}^\top T^* = \mathbf{1}^\top = \mathbf{1}^\top(rI + (1-r)T^*)$ and the fact that $(rI + (1-r)T^*)^{-1}T^* = T^*(rI + (1-r)T^*)^{-1}$ imply that $\mathbf{1}^\top P(r) = \mathbf{1}^\top$, $r \notin \mathcal{S}$. It is easy to verify $P(r^*) = P^*$ by plugging in the definition of $T^*$. Lemma 1 then implies that $\max(\mathcal{S}) < r^*$ and that $P(r')$ is a stochastic matrix for $r' \geq r^*$. To prove the last statement of the theorem we rewrite $P(r)$ by plugging in the definition of $T^*$:

$$P(r) = \frac{r^*}{r}(I - (1 - r^*/r)P^*)^{-1}P^* \tag{6}$$

and consider its first-order derivative w.r.t. $r$:

$$\frac{\partial P(r)}{\partial r} = -\left(\frac{r}{r^*}I + \left(1 - \frac{r}{r^*}\right)P^*\right)^{-2}\frac{(I - P^*)P^*}{r^*}, \tag{7}$$

which exists for $r \in (0, 1] \setminus \mathcal{S}$. By assumption we have $P_{ij}^* = 0$, and by ergodicity of $P^*$ we can assume $(P^*)_{ij}^2 > 0$ (otherwise there exists $k \neq j$ such that $P_{ik}^* = 0$ and $(P^*)_{ik}^2 > 0$). Then we have

$$\left.\frac{\partial P(r)_{ij}}{\partial r}\right|_{r=r^*} = \frac{(P^*)_{ij}^2}{r^*} > 0, \tag{8}$$

implying that there exists $c > 0$ such that for $r \in [r^* - c, r^*)$, $P(r)_{ij} < P_{ij}^* = 0$. This and Lemma 1 then imply the last statement of the theorem.

## D   Sample Complexity Analysis

The analyses here mostly follow those in [1]. Let $O$ denote the observation matrix, which can be the $T$ matrix in First-order Markov models, the $U$ matrix or the product $UT$ in Hidden Markov Models. Define

$$\widetilde{O} := O\mathsf{diag}([\sqrt{\pi_1} \ \sqrt{\pi_2} \ \cdots \ \sqrt{\pi_k}]),$$

$$M_2 := O\mathsf{diag}(\boldsymbol{\pi})O^\top = \widetilde{O}\widetilde{O}^\top \quad \text{and} \quad M_3 := \sum_{i=1}^{k} \pi_i O_i \otimes O_i \otimes O_i.$$

Let $\pi_{\min} := \min_i \pi_i$. We have

$$\sigma_k(O)\sqrt{\pi_{\min}} \leq \sigma_k(\widetilde{O}), \tag{9}$$
$$\sigma_1(\widetilde{O}) \leq \sigma_1(O), \tag{10}$$

where $\sigma_j(\cdot)$ denotes the $j$th largest singular value.

Denote by $\|\cdot\|$ the spectral norm of a matrix or the operator norm of a symmetric third-order tensor induced by the vector 2-norm:

$$\|M\| := \sup_{\|\boldsymbol{\theta}\|_2=1} |M(\boldsymbol{\theta}, \boldsymbol{\theta}, \boldsymbol{\theta})|. \tag{11}$$

Suppose

$$\|\widehat{M_2} - M_2\| = E_2, \tag{12}$$
$$\|\widehat{M_3} - M_3\| \leq E_3, \tag{13}$$

for some $E_2$ and $E_3$ to be determined.

### D.1 Perturbation Lemmas

Let $\widehat{M}_{2,k}$ be the best rank $k$ approximation to $\widehat{M}_2$ in terms of the matrix 2-norm. According to Algorithm A.1, we have

$$\widehat{W}^\top \widehat{M}_{2,k}\widehat{W} = I. \tag{14}$$

Let

$$\widehat{W}^\top M_2 \widehat{W} = ADA^\top \tag{15}$$

be an SVD of $\widehat{W}^\top M_2 \widehat{W}$, where $A \in \mathcal{R}^{k \times k}$. Define

$$W := \widehat{W}AD^{-1/2}A^\top \tag{16}$$

and notice that

$$W^\top M_2 W = AD^{-1/2}A^\top \widehat{W}^\top M_2 \widehat{W}AD^{-1/2}A^\top = I. \tag{17}$$

Let $Q := W^\top \widetilde{O}$ and $\widehat{Q} := \widehat{W}^\top \widetilde{O}$.

**Lemma 2.** *(Lemma C.1 of [1]) Let $\Pi_W$ be the orthogonal projection onto the range of $W$ and $\Pi$ be the orthogonal projection onto the range of $O$. Suppose $E_2 \leq \sigma_k(M_2)/2$. We have the following:*

$$
\begin{aligned}
\|Q\| &= 1, \\
\|\widehat{Q}\| &\leq 2, \\
\|\widehat{W}\| &\leq \frac{2}{\sigma_k(\widetilde{O})}, \\
\|\widehat{W}^\dagger\| &\leq 2\sigma_1(\widetilde{O}), \\
\|W^\dagger\| &\leq 3\sigma_1(\widetilde{O}), \\
\|Q - \widehat{Q}\| &\leq \frac{4E_2}{\sigma_k(\widetilde{O})^2}, \\
\|\widehat{W}^\dagger - W^\dagger\| &\leq \frac{6\sigma_1(\widetilde{O})E_2}{\sigma_k(\widetilde{O})^2}, \\
\|\Pi - \Pi_W\| &\leq \frac{4E_2}{\sigma_k(\widetilde{O})^2}.
\end{aligned}
$$

**Lemma 3.** *Weyl's Theorem. (Theorem 4.11, p.204 in [4]). Let $A, E \in \mathcal{R}^{m \times n}$ with $m \geq n$ be given. Then*

$$\max_{1 \leq i \leq n} |\sigma_i(A + E) - \sigma_i(A)| \leq \|E\|.$$

### D.2 Reconstruction Accuracy

Throughout this section we assume that the number of iterations $\mathsf{N}$ and $\mathsf{L}$ for Algorithm A.2 satisfy the conditions in Theorem A.1.

**Lemma 4.** *Suppose $\max(E_2, E_3) \leq \sigma_k(M_2)/2$. For any $\eta \in (0,1)$, with probability at least $1 - \eta$ the following holds:*

$$\|O - (\widehat{W}^\top)^\dagger \widehat{V}\widehat{\Lambda}\| \leq c\frac{\max(\sigma_1(O), 1)}{\pi_{\min}^{3/2}\min(\sigma_k(O)^2, 1)}\max(E_2, E_3)$$

*for some constant $c > 0$.*

*Proof.* By Theorem A.1, the following hold with probability at least $1 - \eta$:

$$\|V - \widehat{V}\|_F = \sqrt{\sum_i \|V_i - \widehat{V}_i\|^2} \leq \sqrt{\sum_i (64E_3^2)/(1/\sqrt{\pi_{\min}})^2} = 8E_3, \tag{18}$$

$$\|\widehat{\Lambda}\| = \max_i \widehat{1/\sqrt{\pi_i}} \leq \max_i(1/\sqrt{\pi_i} + 5E_3) \leq \pi_{\min}^{-1/2} + 5E_3. \tag{19}$$

With the above two bounds and Lemma 2 we have

$$\|O - (\widehat{W}^\top)^\dagger \widehat{V}\widehat{\Lambda}\| \leq \|O - \Pi_W O\| + \|\Pi_W O - (\widehat{W}^\top)^\dagger \widehat{V}\widehat{\Lambda}\|$$

$$=\|\Pi O - \Pi_W O\| + \|(W^\dagger)^\top V \Lambda - (\widehat{W}^\dagger)^\top \widehat{V}\widehat{\Lambda}\|$$

$$\leq \|\Pi - \Pi_W\|\|O\| + \|(W^\dagger)^\top V \Lambda - (W^\dagger)^\top V\widehat{\Lambda}\| + \|(W^\dagger)^\top V\widehat{\Lambda} - (\widehat{W}^\dagger)^\top \widehat{V}\widehat{\Lambda}\|$$

$$\leq \|\Pi - \Pi_W\| + \|W^\dagger\|\|V\|\|\Lambda - \widehat{\Lambda}\| + \|(W^\dagger)^\top V\widehat{\Lambda} - (W^\dagger)^\top \widehat{V}\widehat{\Lambda}\| + \|(W^\dagger)^\top V\widehat{\Lambda} - (\widehat{W}^\dagger)^\top \widehat{V}\widehat{\Lambda}\|$$

$$\leq \|\Pi - \Pi_W\| + \|W^\dagger\|E_3 + \|W^\dagger\|\|V - \widehat{V}\|\|\widehat{\Lambda}\| + \|W^\dagger - \widehat{W}^\dagger\|\|\widehat{V}\|\|\widehat{\Lambda}\|$$

$$\leq \frac{4E_2}{\sigma_k(\widetilde{O})^2} + 3\sigma_1(\widetilde{O})E_3 + 3\sigma_1(\widetilde{O})\|V - \widehat{V}\|_F\|\widehat{\Lambda}\| + \frac{6\sigma_1(\widetilde{O})E_2}{\sigma_k(\widetilde{O})^2}(\|\widehat{V} - V\|_F + 1)\|\widehat{\Lambda}\|$$

$$\leq c\left(\left(\frac{24}{\sqrt{\pi_{\min}}} + 3\right)\sigma_1(O)E_3 + \left(4 + \frac{6\sigma_1(O)}{\sqrt{\pi_{\min}}}\right)\frac{E_2}{\sigma_k(O)^2\pi_{\min}}\right)$$

$$\leq c\left(\frac{27\sigma_1(O)}{\sqrt{\pi_{\min}}} + \frac{10\max(\sigma_1(O),1)}{\pi_{\min}^{3/2}\sigma_k(O)^2}\right)\max(E_2, E_3)$$

$$\leq c\frac{37\max(\sigma_1(O),1)}{\pi_{\min}^{3/2}\min(\sigma_k(O)^2, 1)}\max(E_2, E_3)$$

where $c > 0$ is a constant large enough to dominate low-order terms like $E_2 E_3$. $\qquad\square$

**Lemma 5.** *With a slight abuse of notation, let $U$ denote a column permutation of the true $U$, $UT$ denote a column permutation of the true $UT$, and $P$ denote a column-and-row permutation of the true $P$, where the permutations involved are the same. Suppose*

$$\max(\|U - \widehat{U}\|, \|\widehat{UT} - UT\|) \leq \sigma_k(rU + (1-r)UT)/2.$$

*We then have*

$$\|P - (r\widehat{U} + (1-r)\widehat{UT})^\dagger\widehat{UT}\| \leq \frac{6\sigma_1(UT)}{\sigma_k(rU + (1-r)UT)^2}\max(\|U - \widehat{U}\|, \|UT - \widehat{UT}\|).$$

*Proof.* First notice that

$$(rU + (1-r)UT)^\dagger(UT)$$

$$=\left((rI + (1-r)T)^\top U^\top U(rI + (1-r)T)\right)^{-1}(rI + (1-r)T)^\top U^\top UT$$

$$=(rI + (1-r)T)^{-1}T = P.$$

Then we have

$$\|P - (r\widehat{U} + (1-r)\widehat{UT})^\dagger\widehat{UT}\| = \|(rU + (1-r)(UT))^\dagger UT - (r\widehat{U} + (1-r)\widehat{UT})^\dagger\widehat{UT}\|$$

$$\leq \|(rU + (1-r)UT)^\dagger(UT) - (r\widehat{U} + (1-r)\widehat{UT})^\dagger(UT)\| +$$

$$\quad \|(r\widehat{U} + (1-r)\widehat{UT})^\dagger(UT) - (r\widehat{U} + (1-r)\widehat{UT})^\dagger\widehat{UT}\|$$

$$\leq \|(rU + (1-r)UT)^\dagger - (r\widehat{U} + (1-r)\widehat{UT})^\dagger\|\|UT\| + \|(r\widehat{U} + (1-r)\widehat{UT})^\dagger\|\|UT - \widehat{UT}\|. \tag{20}$$

By Lemma 3 and the assumption of the lemma, we have

$$\sigma_k(rU + (1-r)UT)/2 \leq \sigma_k(r\widehat{U} + (1-r)\widehat{UT}) \leq 3\sigma_k(rU + (1-r)UT)/2,$$

showing that $\text{rank}(r\widehat{U} + (1-r)\widehat{UT}) = k$ and

$$\|(r\widehat{U} + (1-r)\widehat{UT})^\dagger\| = 1/\sigma_k(r\widehat{U} + (1-r)\widehat{UT}) \leq 2/\sigma_k(rU + (1-r)UT).$$

Because $\text{rank}(r\widehat{U} + (1-r)\widehat{UT}) = \text{rank}(rU + (1-r)UT) = k$, Theorem 3.4 in [3] indicates that

$$\|(rU + (1-r)UT)^\dagger - (r\widehat{U} + (1-r)\widehat{UT})^\dagger\|$$

$$\leq \sqrt{2}\|(rU + (1-r)UT)^\dagger\|\|(r\widehat{U} + (1-r)\widehat{UT})^\dagger\|\|r(U - \widehat{U}) + (1-r)(UT - \widehat{UT})\|$$

$$\leq \frac{\sqrt{2}(r\|U - \widehat{U}\| + (1-r)\|\widehat{UT} - UT\|)}{\sigma_k(rU + (1-r)UT)\sigma_k(r\widehat{U} + (1-r)\widehat{UT})} \leq \frac{2\sqrt{2}(r\|U - \widehat{U}\| + (1-r)\|UT - \widehat{UT}\|)}{\sigma_k(rU + (1-r)UT)^2}.$$

Applying these bounds to (20) then leads to

$$\|P - (r\widehat{U} + (1-r)\widehat{UT})^{\dagger}\widehat{UT}\|$$

$$\leq \frac{2\sqrt{2}\sigma_1(UT)\left(r\|U - \widehat{U}\| + (1-r)\|UT - \widehat{UT}\|\right)}{\sigma_k(rU + (1-r)UT)^2} + \frac{2\|UT - \widehat{UT}\|}{\sigma_k(rU + (1-r)UT)}$$

$$= \frac{r2\sqrt{2}\sigma_1(UT)\|U - \widehat{U}\|}{\sigma_k(rU + (1-r)UT)^2} + \frac{\left((1-r)2\sqrt{2}\sigma_1(UT) + 2\sigma_k(rU + (1-r)UT)\right)\|UT - \widehat{UT}\|}{\sigma_k(rU + (1-r)UT)^2}$$

$$\leq \frac{\max(r2\sqrt{2}, (1-r)2\sqrt{2} + 2)\sigma_1(UT)}{\sigma_k(rU + (1-r)UT)^2} \max(\|U - \widehat{U}\|, \|UT - \widehat{UT}\|)$$

$$\leq \frac{6\sigma_1(UT)}{\sigma_k(rU + (1-r)UT)^2} \max(\|U - \widehat{U}\|, \|UT - \widehat{UT}\|),$$

in which we use the fact $\sigma_1(UT) \geq \sigma_1(rU + (1-r)UT) \geq \sigma_k(rU + (1-r)UT)$. $\qquad\square$

### D.3 Concentration of empirical averages

**Lemma 6.** *Let $\{\mathbf{y}_i\}_{i=1}^N$ be $N$ i.i.d. random vectors in $\mathcal{R}^m$. Let $\boldsymbol{\mu} := \mathbb{E}[\mathbf{y}_i], \Sigma := Var(\mathbf{y}_i)$ and $\sigma_{\max}^2 := \max_d \Sigma_{dd}$. Let $\bar{\boldsymbol{\mu}} := (\sum_i \mathbf{y}_i)/N$. Then*

$$Prob(\|\bar{\boldsymbol{\mu}} - \boldsymbol{\mu}\|_2 \geq \epsilon) \leq \frac{m\sigma_{\max}^2}{N\epsilon^2}.$$

*Proof.* This lemma is a straightforward consequence of the Markov inequality:

$$\text{Prob}(\|\bar{\boldsymbol{\mu}} - \boldsymbol{\mu}\|_2 \geq \epsilon) \quad = \quad \text{Prob}(\|\bar{\boldsymbol{\mu}} - \boldsymbol{\mu}\|_2^2 \geq \epsilon^2) \tag{21}$$

$$\leq \quad \frac{\mathbb{E}[\|\bar{\boldsymbol{\mu}} - \boldsymbol{\mu}\|_2^2]}{\epsilon^2} \tag{22}$$

$$= \quad \frac{\sum_d \mathbb{E}(\bar{\boldsymbol{\mu}}_d - \boldsymbol{\mu}_d)^2}{\epsilon^2} \quad = \quad \frac{\text{Tr}(\Sigma)}{N\epsilon^2} \quad \leq \quad \frac{m\sigma_{\max}^2}{N\epsilon^2}. \tag{23}$$

$$\square$$

**Lemma 7.** *Let $\widehat{V}_1, \widehat{V}_2, \widehat{V}_3, \widehat{C}_2, \widehat{C}_3$ denote averages of $N$ independent draws of $\mathbf{x}_1, \mathbf{x}_1\mathbf{x}_1^{\top}, \mathbf{x}_1 \otimes \mathbf{x}_1 \otimes \mathbf{x}_1, \mathbf{x}_1\mathbf{x}_2^{\top}, \mathbf{x}_1 \otimes \mathbf{x}_2 \otimes \mathbf{x}_3$ from the generative process in Section 3.2. Let $u_{\max} := \max_{i,j} |U_{ij}|$. Then*

$$Prob(\|\widehat{V}_1 - V_1\|_2 \geq \epsilon) \quad \leq \quad \frac{m(u_{\max}^2 + \sigma^2)}{N\epsilon^2}, \tag{24}$$

$$Prob(\|\widehat{V}_2 - V_2\|_F \geq \epsilon) \quad \leq \quad \frac{m^2(u_{\max}^2 + \sigma^2)^2}{N\epsilon^2}, \tag{25}$$

$$Prob(\|\widehat{V}_3 - V_3\|_F \geq \epsilon) \quad \leq \quad \frac{m^3(u_{\max}^2 + \sigma^2)^3}{N\epsilon^2}, \tag{26}$$

$$Prob(\|\widehat{C}_2 - C_2\|_F \geq \epsilon) \quad \leq \quad \frac{m^2(u_{\max}^2 + \sigma^2)^2}{N\epsilon^2}, \tag{27}$$

$$Prob(\|\widehat{C}_3 - C_3\|_F \geq \epsilon) \quad \leq \quad \frac{m^3(u_{\max}^2 + \sigma^2)^3}{N\epsilon^2}. \tag{28}$$

*Proof.* Based on Lemma 6, it suffices to bound $\sigma_{\max}^2$ in these five cases:

$$\max_i \mathrm{Var}((\mathbf{x}_1)_i) \;\leq\; \max_i \mathbb{E}[(\mathbf{x}_1)_i^2] \;=\; \max_i \mathbb{E}_{\mathbf{h}_1}[\sigma^2 + (U\mathbf{h}_1)_i^2] \;\leq\; \sigma^2 + \max_{i,k} U_{ik}^2, \quad (29)$$

$$\begin{aligned}
\max_{i,j} \mathrm{Var}((\mathbf{x}_1)_i(\mathbf{x}_1)_j) &\leq \max_{i,j} \mathbb{E}[(\mathbf{x}_1)_i^2(\mathbf{x}_1)_j^2] = \max_{i,j} \mathbb{E}_{\mathbf{h}_1}[(\sigma^2 + (U\mathbf{h}_1)_i^2)(\sigma^2 + (U\mathbf{h}_1)_j^2)] \\
&\leq \max_{i,j,l}(\sigma^2 + U_{il}^2)(\sigma^2 + U_{jl}^2) \;\leq\; (\sigma^2 + \max_{i,j} U_{ij}^2)^2, \quad (30)
\end{aligned}$$

$$\begin{aligned}
\max_{i,j} \mathrm{Var}((\mathbf{x}_1)_i(\mathbf{x}_2)_j) &\leq \max_{i,j} \mathbb{E}[(\mathbf{x}_1)_i^2(\mathbf{x}_2)_j^2] = \max_{i,j} \mathbb{E}_{\boldsymbol{\pi}_0}\big[\mathbb{E}[(\mathbf{x}_1)_i^2|\boldsymbol{\pi}_0]\mathbb{E}[(\mathbf{x}_2)_i^2|\boldsymbol{\pi}_0]\big] \quad (31) \\
&\leq \max_{i,j} \sup_{\boldsymbol{\pi}_0} \mathbb{E}[(\mathbf{x}_1)_i^2|\boldsymbol{\pi}_0]\mathbb{E}[(\mathbf{x}_2)_i^2|\boldsymbol{\pi}_0] \;\leq\; \big(\max_i \sup_{\boldsymbol{\pi}_0} \mathbb{E}[(\mathbf{x}_1)_i^2|\boldsymbol{\pi}_0]\big)^2 \quad (32) \\
&= \big(\max_i \sup_{\boldsymbol{\pi}_0} \sum_k U_{ij}^2(T\boldsymbol{\pi}_0)_k + \sigma^2\big)^2 \quad (33) \\
&= \big(\max_i \max_{j'} \sum_k U_{ij}^2 T_{jj'} + \sigma^2\big)^2 \;\leq\; (\max_{i,j} U_{ij}^2 + \sigma^2)^2. \quad (34)
\end{aligned}$$

With similar arguments, we have that

$$\max_{i,j,l} \mathrm{Var}((\mathbf{x}_1)_i(\mathbf{x}_1)_j(\mathbf{x}_1)_l) \;\leq\; (\max_{i,j} U_{ij}^2 + \sigma^2)^3, \quad (35)$$

$$\max_{i,j,l} \mathrm{Var}((\mathbf{x}_1)_i(\mathbf{x}_2)_j(\mathbf{x}_3)_l) \;\leq\; (\max_{i,j} U_{ij}^2 + \sigma^2)^3. \quad (36)$$

$\square$

**Lemma 8.** *Let $\widehat{M}_2, \widehat{M}_3, \widehat{M'}_2, \widehat{M'}_3$ denote estimates obtained by plugging in empirical averages of independent samples as in Lemma 7 and $\widehat{\sigma_2} := \lambda_{\min}(\widehat{V_2} - \widehat{V_1}\widehat{V_1}^\top)$, where $\lambda_{\min}(\cdot)$ denotes the smallest eigenvalue in modulus. Define $\nu := \max(\sigma^2 + u_{\max}^2, 1)$. We then have the following:*

$$\begin{aligned}
Prob(\|M_2 - \widehat{M}_2\| \geq \epsilon) &\leq \frac{75m^2\nu^2}{N\epsilon^2}, \\
Prob(\|M_3 - \widehat{M}_3\| \geq \epsilon) &\leq \frac{1000m^4\nu^3}{N\epsilon^2}, \\
Prob(\|M'_2 - \widehat{M'_2}\| \geq \epsilon) &\leq \frac{50(\alpha_0 + 1)^2 m^2\nu^2}{N\epsilon^2}, \\
Prob(\|M'_3 - \widehat{M'_3}\| \geq \epsilon) &\leq \frac{1100k^2m^3(\alpha_0 + 2)^2(\alpha_0 + 1)^2\nu^3}{N\epsilon^2}.
\end{aligned}$$

*Proof.* We first note that it is easy verify $\mathbf{z}^\top(\widehat{V_2} - \widehat{V_1}\widehat{V_1}^\top)\mathbf{z} \geq 0$ for any real vector $\mathbf{z}$, so $\widehat{\sigma^2}$ is always non-negative. By Lemma 3, we have

$$\begin{aligned}
|\sigma^2 - \widehat{\sigma^2}| &\leq \|V_2 - V_1 V_1^\top - (\widehat{V_2} - \widehat{V_1}\widehat{V_1}^\top)\| \;\leq\; \|V_2 - \widehat{V_2}\| + \|V_1 V_1^\top - \widehat{V_1}\widehat{V_1}^\top\| \\
&\leq \|V_2 - \widehat{V_2}\| + \|\widehat{V_1} - V_1\|(\|\widehat{V_1}\| + \|V_1\|) \\
&\leq \|V_2 - \widehat{V_2}\| + 2\|V_1\|\|\widehat{V_1} - V_1\| + \|V_1 - \widehat{V_1}\|^2.
\end{aligned}$$

We also need the following

$$\|V_1\|^2 \;=\; \|U\boldsymbol{\pi}\|^2 \;=\; \sum_i \left(\sum_j U_{ij}\pi_j\right)^2 \;\leq\; \sum_{i,j} \pi_j U_{ij}^2 \;\leq\; \sum_i \max_j U_{ij}^2 \;\leq\; m u_{\max}^2.$$

Then we have

$$\begin{aligned}
\|\widehat{M}_2 - M_2\| &\leq \|\widehat{V_2} - V_2\| + |\widehat{\sigma^2} - \sigma^2| \\
&\leq 2\|\widehat{V_2} - V_2\| + 2\|V_1\|\|\widehat{V_1} - V_1\| + \|\widehat{V_1} - V_1\|^2 \\
&\leq 2\|\widehat{V_2} - V_2\|_F + 2\|V_1\|\|\widehat{V_1} - V_1\| + \|\widehat{V_1} - V_1\|^2,
\end{aligned}$$

which implies

$$\text{Prob}(\|\widehat{M_2} - M_2\| \geq \epsilon)$$
$$\leq \quad \text{Prob}(2\|\widehat{V_2} - V_2\|_F + 2\|V_1\|\|\widehat{V_1} - V_1\| + \|\widehat{V_1} - V_1\|^2 \geq \epsilon)$$
$$\leq \quad \text{Prob}(2\|\widehat{V_2} - V_2\|_F \geq \epsilon/3) + \text{Prob}(2\|V_1\|\|\widehat{V_1} - V_1\| \geq \epsilon/3) + \text{Prob}(\|\widehat{V_1} - V_1\|^2 \geq \epsilon/3)$$
$$\leq \quad \frac{36m^2(u_{\max}^2 + \sigma^2)^2}{N\epsilon^2} + \frac{36\|V_1\|^2 m(u_{\max}^2 + \sigma^2)}{N\epsilon^2} + \frac{3m(u_{\max}^2 + \sigma^2)}{N\epsilon}$$
$$\leq \quad \frac{36m^2(u_{\max}^2 + \sigma^2)^2}{N\epsilon^2} + \frac{36m^2 u_{\max}^2(u_{\max}^2 + \sigma^2)}{N\epsilon^2} + \frac{3m(u_{\max}^2 + \sigma^2)}{N\epsilon}$$
$$\leq \quad \frac{75m^2(u_{\max}^2 + \sigma^2)^2}{N\epsilon^2}.$$

Similarly, we have

$$\|M_3 - \widehat{M_3}\| \quad \leq \quad \|V_3 - \widehat{V_3}\|_F + 3\|V_1 \otimes_1 (\sigma^2 I) - \widehat{V_1} \otimes_1 (\widehat{\sigma^2} I)\|_F$$
$$= \quad \|V_3 - \widehat{V_3}\|_F + 3\sqrt{m}\|\sigma^2 V_1 - \widehat{\sigma^2}\widehat{V_1}\|$$
$$\leq \quad \|V_3 - \widehat{V_3}\|_F + 3\sqrt{m}(\sigma^2\|V_1 - \widehat{V_1}\| + |\sigma^2 - \widehat{\sigma^2}|(\|V_1\| + \|\widehat{V_1} - V_1\|))$$
$$\leq \quad \|V_3 - \widehat{V_3}\|_F + \|V_1 - \widehat{V_1}\|3\sqrt{m}(\sigma^2 + 2mu_{\max}^2) + \|V_2 - \widehat{V_2}\|3u_{\max}m$$
$$+ \|V_1 - \widehat{V_1}\|^2 9u_{\max}m + 3\sqrt{m}(\|V_1 - \widehat{V_1}\|\|\widehat{V_2} - V_2\| + \|V_1 - \widehat{V_1}\|^3),$$

implying

$$\text{Prob}(\|M_3 - \widehat{M_3}\| \geq \epsilon)$$
$$\leq \quad \text{Prob}(\|V_3 - \widehat{V_3}\|_F \geq \epsilon/6) + \text{Prob}(\|V_1 - \widehat{V_1}\| \geq \epsilon/(18\sqrt{m}(\sigma^2 + 2mu_{\max}^2)))$$
$$+ \text{Prob}(\|V_2 - \widehat{V_2}\| \geq \epsilon/(18u_{\max}m)) + \text{Prob}(\|V_1 - \widehat{V_1}\|^2 \geq \epsilon/(54u_{\max}m))$$
$$+ \text{Prob}\left(\|V_1 - \widehat{V_1}\| \geq \sqrt{\epsilon/(18\sqrt{m})}\right) + \text{Prob}\left(\|\|V_2 - \widehat{V_2}\| \geq \sqrt{\epsilon/(18\sqrt{m})}\right)$$
$$+ \text{Prob}(\|V_1 - \widehat{V_1}\|^3 \geq \epsilon/(18\sqrt{m}))$$
$$\leq \quad \frac{36m^3(u_{\max}^2 + \sigma^2)^3}{N\epsilon^2} + \frac{324m^2(\sigma^2 + 2mu_{\max}^2)^2(\sigma^2 + u_{\max}^2)}{N\epsilon^2} + \frac{324u_{\max}^2 m^4(\sigma^2 + u_{\max}^2)^2}{N\epsilon^2}$$
$$+ \frac{54u_{\max}m^2(\sigma^2 + u_{\max}^2)}{N\epsilon} + \frac{18m^{3/2}(\sigma^2 + u_{\max}^2)}{N\epsilon} + \frac{18m^{5/2}(\sigma^2 + u_{\max}^2)^2}{N\epsilon}$$
$$+ \frac{36^{1/3}m^{4/3}(\sigma^2 + u_{\max}^2)}{N\epsilon^{2/3}}$$
$$\leq \quad \frac{1000m^4(\max(\sigma^2 + u_{\max}^2, 1))^3}{N\epsilon^2}.$$

Using similar arguments, we have

$$\|M_2' - \widehat{M_2'}\| \quad \leq \quad (\alpha_0 + 1)\|C_2 - \widehat{C_2}\|_F + \alpha_0\|V_1 V_1^\top - \widehat{V_1}\widehat{V_1}^\top\|_F$$
$$\leq \quad (\alpha_0 + 1)\|C_2 - \widehat{C_2}\|_F + 2\alpha_0\|V_1\|\|\widehat{V_1} - V_1\| + \alpha_0\|\widehat{V_1} - V_1\|^2,$$

and therefore

$$\text{Prob}(\|M_2' - \widehat{M_2'}\| \geq \epsilon)$$
$$\leq \quad \text{Prob}(\|C_2 - \widehat{C_2}\|_F \geq \frac{\epsilon}{3(\alpha_0 + 1)}) + \text{Prob}(\|\widehat{V_1} - V_1\| \geq \frac{\epsilon}{6\alpha_0\|V_1\|})$$
$$+ \text{Prob}(\|\widehat{V_1} - V_1\|^2 \geq \frac{\epsilon}{3\alpha_0})$$
$$\leq \quad \frac{9(\alpha_0 + 1)^2 m^2(\sigma^2 + u_{\max}^2)^2}{N\epsilon^2} + \frac{36\alpha_0^2 m^2 u_{\max}^2(\sigma^2 + u_{\max}^2)}{N\epsilon^2} + \frac{3\alpha_0 m(\sigma^2 + u_{\max}^2)}{N\epsilon}$$
$$\leq \quad \frac{50(\alpha_0 + 1)^2 m^2(\sigma^2 + u_{\max}^2)^2}{N\epsilon^2}.$$

Finally, we have

$$\|M'_3 - \widehat{M'}_3\|$$

$$\leq \frac{(\alpha_0 + 2)(\alpha_0 + 1)}{2}\|C_3 - \widehat{C_3}\|_F + \frac{3(\alpha_0 + 1)\alpha_0}{2}\|V_1 \otimes_1 C_2 - \widehat{V_1} \otimes \widehat{C_2}\|_F$$

$$+ \alpha_0^2 \|V_1 \otimes V_1 \otimes V_1 - \widehat{V_1} \otimes \widehat{V_1} \otimes \widehat{V_1}\|_F$$

$$\leq \frac{(\alpha_0 + 2)(\alpha_0 + 1)}{2}\|C_3 - \widehat{C_3}\|_F + \frac{3(\alpha_0 + 1)\alpha_0}{2}\|V_1 - \widehat{V_1}\|\|C_2\|_F + \frac{3(\alpha_0 + 1)\alpha_0}{2}\|\widehat{V_1}\|\|C_2 - \widehat{C_2}\|_F$$

$$+ 3\alpha_0^2\|V_1\|^2\|V_1 - \widehat{V_1}\| + 3\alpha_0^2\|V_1\|\|V_1 - \widehat{V_1}\|^2 + \alpha_0^2\|V_1 - \widehat{V_1}\|^3$$

$$\leq \frac{(\alpha_0 + 2)(\alpha_0 + 1)}{2}\|C_3 - \widehat{C_3}\|_F + \frac{3(\alpha_0 + 1)\alpha_0}{2}\|V_1 - \widehat{V_1}\|\|C_2\|_F + \frac{3(\alpha_0 + 1)\alpha_0}{2}\|V_1\|\|C_2 - \widehat{C_2}\|_F$$

$$+ \frac{3(\alpha_0 + 1)\alpha_0}{2}\|V_1 - \widehat{V_1}\|\|C_2 - \widehat{C_2}\|_F + 3\alpha_0^2\|V_1\|^2\|V_1 - \widehat{V_1}\| + 3\alpha_0^2\|V_1\|\|V_1 - \widehat{V_1}\|^2 + \alpha_0^2\|V_1 - \widehat{V_1}\|^3$$

$$\leq \frac{(\alpha_0 + 2)(\alpha_0 + 1)}{2}\|C_3 - \widehat{C_3}\|_F + 5(\alpha_0 + 1)\alpha_0 k m u_{\max}^2\|V_1 - \widehat{V_1}\| + \frac{3(\alpha_0 + 1)\alpha_0}{2}\|V_1\|\|C_2 - \widehat{C_2}\|_F$$

$$+ \frac{3(\alpha_0 + 1)\alpha_0}{2}\|V_1 - \widehat{V_1}\|\|C_2 - \widehat{C_2}\|_F + 3\alpha_0^2\|V_1\|\|V_1 - \widehat{V_1}\|^2 + \alpha_0^2\|V_1 - \widehat{V_1}\|^3$$

using the fact that

$$\|C_2\|_F = \left\|UT\left(\frac{\mathsf{diag}\boldsymbol{\pi} + \alpha_0\boldsymbol{\pi}\boldsymbol{\pi}^\top}{\alpha_0 + 1}\right)T^\top U^\top\right\|_F \leq \|UT\|_F^2 \leq k m u_{\max}^2,$$

and thus

$$\mathsf{Prob}(\|M'_3 - \widehat{M'}_3\| \geq \epsilon) \leq \mathsf{Prob}\left(\|C_3 - \widehat{C_3}\|_F \geq \frac{\epsilon}{3(\alpha_0 + 2)(\alpha_0 + 1)}\right)$$

$$+ \mathsf{Prob}\left(\|V_1 - \widehat{V_1}\|_F \geq \frac{\epsilon}{30(\alpha_0 + 1)\alpha_0 k m u_{\max}^2}\right) + \mathsf{Prob}\left(\|C_2 - \widehat{C_2}\|_F \geq \frac{\epsilon}{9(\alpha_0 + 1)\alpha_0}\right)$$

$$+ \mathsf{Prob}\left(\|V_1 - \widehat{V_1}\|^2 \geq \frac{\epsilon}{18\alpha_0^2\|V_1\|}\right) + \mathsf{Prob}\left(\|V_1 - \widehat{V_1}\|^3 \geq \frac{\epsilon}{6\alpha_0^2}\right)$$

$$\leq \frac{9m^2(\alpha_0 + 2)^2(\alpha_0 + 1)^2(\sigma^2 + u_{\max}^2)^3}{N\epsilon^2} + \frac{900k^2m^3(\alpha_0 + 1)^2\alpha_0^2 u_{\max}^4(\sigma^2 + u_{\max}^2)}{N\epsilon^2}$$

$$+ \frac{81(\alpha_0 + 1)^2\alpha_0^2 m^2(\sigma^2 + u_{\max}^2)^2}{N\epsilon^2} + \frac{18\alpha_0^2 m^{3/2}u_{\max}(\sigma^2 + u_{\max}^2)}{N\epsilon} + \frac{6m\alpha_0^{4/3}(\sigma^2 + u_{\max}^2)}{N\epsilon^{2/3}}$$

$$\leq \frac{1100k^2m^3(\alpha_0 + 2)^2(\alpha_0 + 1)^2(\sigma^2 + u_{\max}^2)^3}{N\epsilon^2}.$$

□

# E    Proof of Theorem 4

Let $\widehat{U}$ and $\widehat{UT}$ be column-permuted as described in Algorithm 1. Let

$$\delta_{\min} := \min_{i,j}|1/\sqrt{\pi_i} - 1/\sqrt{\pi_j}|.$$

If $\max(E_3, E'_3) \leq \delta_{\min}/15$, Theorem 5.1 of [2] implies that for any $\eta \in (0, 1)$, with probability at least $1 - \eta$, the columns of $\widehat{U}$ and $\widehat{UT}$ are matched to the same permutation of the columns of the true $U$ and $UT$, respectively. As in Lemma 5, let $U, UT$, and $P$ denote proper permutations of the true matrices. We then have

$$\mathsf{Prob}\left(\max(\|U - \widehat{U}\|, \|UT - \widehat{UT}\|) \geq \frac{\epsilon\sigma_k(rU + (1-r)UT)^2}{6\sigma_1(UT)}\right)$$

$$\leq \mathsf{Prob}\left(\|U - \widehat{U}\| \geq \frac{\epsilon\sigma_k(rU + (1-r)UT)^2}{6\sigma_1(UT)}\right) + \mathsf{Prob}\left(\|UT - \widehat{UT}\| \geq \frac{\epsilon\sigma_k(rU + (1-r)UT)^2}{6\sigma_1(UT)}\right).$$

Let the failure probability for the tensor decomposition method be set to $\frac{\eta}{4}$. Then by Lemma 4 we can bound the first term as follows:

$$\text{Prob}\left(\|U - \widehat{U}\| \geq \frac{\epsilon\sigma_k(rU + (1-r)UT)^2}{6\sigma_1(UT)}\right)$$

$$\leq \text{Prob}\left(\max(E_2, E_3) \geq \frac{\epsilon\sigma_k(rU + (1-r)UT)^2\pi_{\min}^{3/2}\min(\sigma_k(U)^2, 1)}{6\sigma_1(UT)c\max(\sigma_1(U), 1)}\right)$$

$$+ \text{Prob}(\max(E_2, E_3) \geq \sigma_k(M_2)/2) + \frac{\eta}{4} + \text{Prob}(E_3 \geq \delta_{\min}/15),$$

where the first term in the r.h.s is based on Lemma 4 conditioned on the event that $\max(E_2, E_3) \geq \sigma_k(M_2)/2$ and the tensor decomposition method succeeds, the second and the third terms bound the probability that the event does not occur, and the last term bounds the probability of incorrectly matching the columns of $\widehat{U}$ and $U$. To continue the bound we use Lemma 8 to have

$$\text{Prob}\left(\max(E_2, E_3) \geq \frac{\epsilon\sigma_k(rU + (1-r)UT)^2\pi_{\min}^{3/2}\min(\sigma_k(U)^2, 1)}{6\sigma_1(UT)c\max(\sigma_1(U), 1)}\right)$$

$$\leq \frac{(2700m^2\nu^2 + 36000m^4\nu^3)\sigma_1(UT)^2c^2\max(\sigma_1(U)^2, 1)}{N\epsilon^2\sigma_k(rU + (1-r)UT)^4\pi_{\min}^3\min(\sigma_k(U)^4, 1)}$$

$$\leq \frac{39000m^4\nu^3\sigma_1(UT)^2c^2\max(\sigma_1(U)^2, 1)}{N\epsilon^2\sigma_k(rU + (1-r)UT)^4\pi_{\min}^3\min(\sigma_k(U)^4, 1)},$$

$$\text{Prob}(\max(E_2, E_3) \geq \sigma_k(M_2)/2) \leq \frac{300m^2\nu^2 + 4000m^4\nu^3}{N\sigma_k(M_2)^2} \leq \frac{4300m^4\nu^3}{N\sigma_k(M_2)^2},$$

$$\text{Prob}(E_3 \geq \delta_{\min}/15) \leq \frac{225000m^4\nu^3}{N\delta_{\min}^2}.$$

Thus, by setting the sample size $N$ so that

$$N \geq \frac{12m^4\nu^3}{\eta}\max\left(\frac{225000}{\delta_{\min}^2}, \frac{4300}{\sigma_k(M_2)^2}, \frac{39000\sigma_1(UT)^2c^2\max(\sigma_1(U)^2, 1)}{\epsilon^2\sigma_k(rU + (1-r)UT)^4\pi_{\min}^3\min(\sigma_k(U)^4, 1)}\right),$$

we have

$$\text{Prob}\left(\|U - \widehat{U}\| \geq \frac{\epsilon\sigma_k(rU + (1-r)UT)^2}{6\sigma_1(UT)}\right) \leq \frac{\eta}{2}, \tag{37}$$

where the randomness is from both the data and the algorithm. Using similar arguments, we have that for sample size $N$ such that

$$N \geq \frac{12k^2m^3(\alpha_0 + 2)^2(\alpha_0 + 1)^2\nu^3}{\eta} \cdot$$

$$\max\left(\frac{225000}{\delta_{\min}^2}, \frac{4600}{\sigma_k(M'_2)^2}, \frac{42000\sigma_1(UT)^2(c')^2\max(\sigma_1(UT)^2, 1)}{\epsilon^2\sigma_k(rU + (1-r)UT)^4\pi_{\min}^3\min(\sigma_k(UT)^4, 1)}\right),$$

the following holds:

$$\text{Prob}\left(\|UT - \widehat{UT}\| \geq \frac{\epsilon\sigma_k(rU + (1-r)UT)^2}{6\sigma_1(UT)}\right) \leq \frac{\eta}{2}.$$

Combining the two bounds (37) and (E), we have for

$$N \geq \frac{12\max(k^2, m)m^3\nu^3(\alpha_0 + 2)^2(\alpha_0 + 1)^2}{\eta} \cdot$$

$$\max\left(\frac{225000}{\delta_{\min}^2}, \frac{4600}{\min(\sigma_k(M'_2), \sigma_k(M_2))^2}, \frac{42000c^2\sigma_1(UT)^2\max(\sigma_1(UT), \sigma_1(U), 1)^2}{\epsilon^2\sigma_k(rU + (1-r)UT)^4\min(\sigma_k(UT), \sigma_k(U), 1)^4}\right),$$

the following bound holds for any $\epsilon > 0$ and $\eta \in (0, 1)$:

$$\text{Prob}\left(\max(\|U - \widehat{U}\|, \|UT - \widehat{UT}\|) \leq \frac{\epsilon\sigma_k(rU + (1-r)UT)^2}{6\sigma_1(UT)}\right) \geq 1 - \eta,$$

which by Lemma 5 implies that

$$\text{Prob}(\|P - (r\widehat{U} + (1-r)\widehat{UT})^\dagger\widehat{UT}\| \leq \epsilon) \geq 1 - \eta.$$

## F  Comparison between Proposed Method and Variational EM

For the purpose of comparison, we derive here a variational EM algorithm for learning HMMs from non-sequence data. First notice that the generative process in Section 3.2 specifies only the variance rather than an entire distribution for the observation model. To make it easier to derive a maximum-likelihood based algorithm, here we make an extra assumption of the observation model being Gaussian. The full joint probability of data and latent variables then takes the following form:

$$
f(\{\mathbf{x}_i^j\}, \{\mathbf{h}_i^j\}, \{t_i^j\}, \{\mathbf{s}_i^j\}, \{\boldsymbol{\pi}_0^j\} \mid U, \sigma^2, P, r, \boldsymbol{\alpha})
$$
$$
= \prod_{j=1}^N \left( \prod_{i=1}^n \Big( \prod_{l=1}^k \mathcal{N}(\mathbf{x}_i^j \mid U_l, \sigma^2 I)^{\mathbf{h}_{il}^j} \Big) \Big( \prod_{l',l} ((P^{t_i^j})_{l'l})^{\mathbf{h}_{il'}^j \mathbf{s}_{il}^j} \Big) \mathsf{Geometric}(t_i^j \mid r) \Big( \prod_l ((\boldsymbol{\pi}_0^j)_l)^{\mathbf{s}_{il}^j} \Big) \right) \cdot
$$
$$
\mathsf{Dirichlet}(\boldsymbol{\pi}_0^j \mid \boldsymbol{\alpha}),
$$
(38)

in which we use super-script as set indices and sub-scripts as data indices within a set wherever appropriate. The goal is to maximize the marginal likelihood of the data w.r.t to the parameters. We begin by marginalizing over the latent times $\{t_i^j\}$:

$$
f(\{\mathbf{x}_i^j\}, \{\mathbf{h}_i^j\}, \{\mathbf{s}_i^j\}, \{\boldsymbol{\pi}_0^j\} \mid U, \sigma^2, T, \boldsymbol{\alpha})
$$
$$
= \prod_{j=1}^N \left( \prod_{i=1}^n \Big( \prod_{l=1}^k \mathcal{N}(\mathbf{x}_i^j \mid U_l, \sigma^2 I)^{\mathbf{h}_{il}^j} \Big) \Big( \prod_{l',l} T_{l'l}^{\mathbf{h}_{il'}^j \mathbf{s}_{il}^j} \Big) \Big( \prod_l ((\boldsymbol{\pi}_0^j)_l)^{\mathbf{s}_{il}^j} \Big) \right) \mathsf{Dirichlet}(\boldsymbol{\pi}_0^j \mid \boldsymbol{\alpha}),
$$
(39)

where $T$ is the expected hidden state transition matrix defined in Theorem 3. As in the tensor factorization approach, we recover $P$ and $r$ from the estimated $T$ using the proposed search heuristics. Because the posterior distribution of the remaining latent variables still leads to an intractable E step, we take the following factorized approximation:

$$
f(\{\mathbf{h}_i^j\}, \{\mathbf{s}_i^j\}, \{\boldsymbol{\pi}_0^j\} \mid \{\mathbf{x}_i^j\}, U, \sigma^2, T, \boldsymbol{\alpha}) \approx q(\{\mathbf{h}_i^j\}, \{\mathbf{s}_i^j\} \mid \{\Phi_j^i\}) q(\{\boldsymbol{\pi}_0^j\} \mid \{\boldsymbol{\beta}^j\}),
$$
(40)

where

$$
q(\{\mathbf{h}_i^j\}, \{\mathbf{s}_i^j\} \mid \{\Phi_i^j\}) \quad := \quad \prod_{i,j,l',l} ((\Phi_i^j)_{l'l})^{\mathbf{h}_{il'}^j \mathbf{s}_{il}^j}, \quad \Phi_i^j \in [0,1]^{k \times k},
$$
(41)

$$
q(\{\boldsymbol{\pi}_0^j\} \mid \{\boldsymbol{\beta}^j\}) \quad := \quad \prod_j \mathsf{Dirichlet}(\boldsymbol{\pi}_0^j \mid \boldsymbol{\beta}^j),
$$
(42)

and obtain the following lower bound on the log marginal likelihood:

$$
g(\{\Phi_i^j\}, \{\boldsymbol{\beta}^j\}, U, \sigma^2, T, \boldsymbol{\alpha})
$$
$$
:= \mathbb{E}_{\{\mathbf{h}_i^j\}, \{\mathbf{s}_i^j\} \mid \{\Phi_i^j\}, \{\boldsymbol{\pi}_0^j\} \mid \{\boldsymbol{\beta}^j\}} \left[ \log \left( \frac{f(\{\mathbf{x}_i^j\}, \{\mathbf{h}_i^j\}, \{\mathbf{s}_i^j\}, \{\boldsymbol{\pi}_0^j\} \mid U, \sigma^2, T, \boldsymbol{\alpha})}{q(\{\mathbf{h}_i^j\}, \{\mathbf{s}_i^j\} \mid \{\Phi_i^j\}) q(\{\boldsymbol{\pi}_0^j\} \mid \{\boldsymbol{\beta}^j\})} \right) \right]
$$
$$
= \mathbb{E}_{\{\mathbf{h}_i^j\}, \{\mathbf{s}_i^j\} \mid \{\Phi_i^j\}, \{\boldsymbol{\pi}_0^j\} \mid \{\boldsymbol{\beta}^j\}} \left[ \log f(\{\mathbf{x}_i^j\}, \{\mathbf{h}_i^j\}, \{\mathbf{s}_i^j\}, \{\boldsymbol{\pi}_0^j\} \mid U, \sigma^2, T, \boldsymbol{\alpha}) \right] -
$$
$$
\mathbb{E}_{\{\mathbf{h}_i^j\}, \{\mathbf{s}_i^j\} \mid \{\Phi_i^j\}} \left[ \log q(\{\mathbf{h}_i^j\}, \{\mathbf{s}_i^j\} \mid \{\Phi_i^j\}) \right] - \mathbb{E}_{\{\boldsymbol{\pi}_0^j\} \mid \{\boldsymbol{\beta}_0^j\}} \left[ \log q(\{\boldsymbol{\pi}_0^j\} \mid \{\boldsymbol{\beta}_0^j\}) \right]
$$
$$
= \sum_{j,i,l,l'} (\Phi_i^j)_{ll'} \left( \log \mathcal{N}(\mathbf{x}_i^j \mid U_l, \sigma^2 I) + \log T_{ll'} \right) + \sum_{j,l} \Big( \sum_{i,l'} (\Phi_i^j)_{l'l} + \alpha_l - 1 \Big) \big( \psi(\boldsymbol{\beta}_l^j) - \psi(\beta_0^j) \big)
$$
$$
- N \Big( \sum_l \log \Gamma(\alpha_l) - \log \Gamma(\alpha_0) \Big) - \sum_{j,i,l,l'} (\Phi_i^j)_{ll'} \log (\Phi_i^j)_{ll'}
$$
$$
- \sum_{j,l} (\boldsymbol{\beta}_l^j - 1)(\psi(\boldsymbol{\beta}_l^j) - \psi(\beta_0^j)) + \sum_j \Big( \sum_l \log \Gamma(\boldsymbol{\beta}_l^j) - \log \Gamma(\beta_0^j) \Big),
$$
(43)

Figure 1: Comparison between Algorithm 1 and Variational EM

where $\psi(\cdot)$ is the digamma function. The variational EM algorithm then amounts to maximizing $g$ iteratively, optimizing over one block of variables at a time while fixing the others:

$$
(\Phi_i^j)_{ll'} \quad \propto \quad \mathcal{N}(\mathbf{x}_i^j \mid U_l, \sigma^2 I) T_{ll'} \exp(\psi(\beta_{l'}^j) - \psi(\beta_0^j)), \tag{44}
$$

$$
(\boldsymbol{\beta}^j)_l \quad = \quad \sum_{i,l'} (\Phi_i^j)_{l'l} + \alpha_l, \tag{45}
$$

$$
U_l \quad := \quad \frac{\sum_{j=1}^N \sum_{i=1}^n \sum_{l'=1}^k (\Phi_i^j)_{ll'} \mathbf{x}_i^j}{\sum_{j=1}^N \sum_{i=1}^n \sum_{l'=1}^k (\Phi_i^j)_{ll'}}, \tag{46}
$$

$$
\sigma^2 \quad := \quad \frac{\sum_{j=1}^N \sum_{i=1}^n \sum_{l,l'} (\Phi_i^j)_{ll'} \|\mathbf{x}_i^j - U_l\|^2}{Nnm}, \tag{47}
$$

$$
T_{ll'} \quad := \quad \frac{\sum_{j=1}^N \sum_{i=1}^n (\Phi_i^j)_{ll'}}{\sum_{j=1}^N \sum_{i=1}^n \sum_{l=1}^k (\Phi_i^j)_{ll'}}, \tag{48}
$$

$$
\boldsymbol{\alpha} \quad := \quad \arg \max_{\{\alpha_l \geq 0\}} \sum_{j=1}^N \sum_{l=1}^k (\alpha_l - 1)(\psi(\beta_l^j) - \psi(\beta_0^j)) - N\Big( \sum_{l=1}^k \log \Gamma(\alpha_l) - \log \Gamma(\alpha_0) \Big) \tag{49}
$$

The update for $\boldsymbol{\alpha}$ is a convex optimization problem whose inverse Hessian can be computed in linear time. We iterate these updates until the lower bound stops increasing.

Finally, we have to match the columns of $U$ with the columns of $T$. Note that the updates imply that the columns of $U$ are aligned with the rows of $T$, so it suffices to match $T$'s rows with its columns. Using the assumptions that $\boldsymbol{\alpha}/\alpha_0 = \boldsymbol{\pi}$ and $\pi_i \neq \pi_j \; \forall \; i \neq j$, we recover the matching by sorting $\boldsymbol{\alpha}/\alpha_0$ and $T\boldsymbol{\alpha}/\alpha_0$.

We compare the proposed method and the above Variational EM algorithm on synthetic data generated from the HMM in Section 4 under the same parameter setting, except that the number of sets $N$ takes smaller values $\{125, 250, 500, 1000, 2000, 4000\}$. We repeat the experiment 20 times with different random draws from the generative process. Figure 1 gives the relative estimation errors for $U$ (in spectral norm) and $P$ (in entrywise 1-norm) for three methods: Algorithm 1 (tensor), variational EM initialized with the output of Algorithm 1 (tensor+vbEM), and variational EM initialized with 100 random parameter values (rand+vbEM). Clearly, Algorithm 1 outperforms the randomly initialized variation EM, and there is barely any improvement resulting from combining the two methods, except when $N$ is very small. In terms of computational efficiency, we observe that Algorithm 1 is orders of magnitude faster than the variational EM algorithm. On our platform with 48 cores (2.3 GHz each) and 512GB of memory, Algorithm 1 takes a couple of hours to finish all 20 experiments, but the variational EM method takes days.

## Footnotes

[1] See Section 4.2 of [2] for the definition of robust eigenvectors/eigenvalues.