[Reviews · NeurIPS 2013]

Submitted by Assigned_Reviewer_1

This paper gives a spectral algorithm for learning HMM from non-sequential observations. Motivated by several scientific examples, the authors define a sampling model for non-sequential observations that shares some similarities with the generative model of Latent Dirichlet Allocation. Then, resorting to recent spectral techniques for learning LDA, HMM, and mixture models, they prove sample bounds for recovering the parameters of an HMM with continuous output from data sampled according to this model. The last section provides a simple simulation that illustrates the behavior of the algorithm in a synthetic example. Proofs of all results are given in the supplementary material.

The main contribution of the paper is to identify a sampling model for non-sequential data generated from an HMM which is amenable to theoretical analysis using recent results on spectral learning, and to prove finite-sample bounds under this model. The experimental evaluation is extremely limited, and perhaps unnecessary given the nature of the paper -- the space used by this part could probably be put to a better use, e.g. highlighting the novel points in the proofs found in the appendix.

In general the paper is well written. The authors explain most of the intuitions behind their results in the text. However, the mathematical style is rather dry -- specially the sample bounds, which favor precision over asymptotic behavior, are quite hard to grasp.

The content of the paper is not extremely original, though there is some novelty in the generative model for non-sequential data and the way it is analyzed. Proof techniques, though involved, look very similar to those in other spectral learning analyses. Besides, the problem of using non-sequence data in a spectral learning algorithm for HMM was already considered in (Spectral Learning of Hidden Markov Models from Dynamic and Static Data, T. Huang and J. Schneider, ICML 2013).

The significance of the method depends on whether the algorithm performs well on real data which, most likely, won't be generated according to the sampling model defined in the paper. The authors leave this as future work, but the sample sizes for which they report good results on a synthetic target suggest that it may not be applicable to problems where only small samples are available.

*** Typo ***
[line 107] V_2 -> X_2
Summary: The paper gives the first finite-sample bounds for learning HMM from non-sequence data from a reasonable sampling model. The interest of this approach depends on whether the algorithm will behave well on the real problems that motivate the model.

Submitted by Assigned_Reviewer_6

This paper presents a tensor factorization approach for parameter
estimation in settings where there is an underlying hidden Markov
model, but we only see a small random fraction of the observations.
The contribution of the paper involves identifying the tensor
factorization structure in the problem (which extends ideas of
Anandkumar et al.), which is relatively straightforward. An
interesting part is showing that you can estimate recover the
transition distribution from an expectation over the sums of the
transition distribution. Sample complexity results and some toy
simulations are provided as well. Overall, a reasonable paper with
some new ideas.

Presentation suggestion: I would suggest defining the actual
model/problem (which is currently in section 3) much earlier - for
example, one shouldn't be subjected to Theorem 1 before even having a
formal definition of what the paper is trying to do. Too much space is
spent reviewing the tensor decomposition framework; a citation to that
work and a brief description of the key ideas suffices, or relegate to
the appendix.

In the definition of the model in section 3.1, please make it explicit
that we are getting $N$ i.i.d. replicates of this model. For a while,
I was confused at how this was even possible if you only have one
random draw from $\pi_0$.

Currently, the theorems very dryly write down true facts, but the text
doesn't really provide any guidance about what are the important
properties to watch out for. For example, line 062 of Appendix A.1
nicely lays out the factorization structure, which I think can be
imported into the main body of the paper. Also, I'd appreciate more
intuition about Theorem 3.

Experiments: it's nice to see some relationship (even if it's just
qualitative) between the empirical findings and the theoretical bounds;
the fact that $U$ is easier to learn is not surprising. I'd be curious
to see how this algorithm compares with EM, since the original
motivation is that EM presumably gets stuck in local optima and this
method does not suffer from that problem. And of course, needless to
say, experiments on real data would be good too, especially taken from
the cited that look at learning from non-sequence data.

318,320: primes should be placed above the subscripts in $M_2$ and $M_3$
Summary: This paper applies recently developed tensor factorization techniques
to the new setting of learning a HMM from non-sequence data. The paper
could be written to convey more insight, but overall it's a reasonable
paper.

Submitted by Assigned_Reviewer_7

- In this paper, a spectral learning based algorithm for learning Markov Model and HMM in a non-sequential setting is derived. The paper deals with learning Markov models and hidden Markov models from sparse realizations obtained at random times.

Proper proofs for empirical moments are given. Also, a sample complexity bound has been provided. The paper is generally well-written and understandable. The comments are as follows:

-Learning a sequential model in a non-sequential setting is not a concept that everybody is familiar of. It may be beneficial to briefly review the existing methods (possibly maximum likelihood based). Author(s) may argue that this can’t be done because of space constraints. However, I believe that, it would be better to motivate the non-sequential sequential model learning rather than to reproduce the tensor decomposition algorithm of Anandkumar et al. I think that the tensor algorithm can be introduced only by giving the general idea of expressing the parameters as symmetric observable tensors. (equation 2) That is, author(s) can exclude the algorithm 2 and theorem 1.

-The proper moment equations for learning a Markov model in a non-sequential setting are defined in page 5. Using these moment equations, it possible to recover the expected transition probability matrix T, and parameters \pi. of the Dirichlet prior, using the symmetric orthogonal tensor decomposition algorithm of Anandkumar et al. from 2012. However, in order to recover the transition matrix itself (P), author(s) propose a search heuristic. In my opinion, this search algorithm can be made clearer to the reader by writing it as a pseudo-code. The assumption of presence of a zero entry in P makes sense if the number of states in the Markov chain is large, which may induce sparsity.
Minor comment: isn’t \pi = \alpha / \alpha_0? Instead of using the proportional (at the beggining of page 5), it is better to give the exact equality to make the reader understand the moment equation proofs more easily.

-Experimental results are limited to synthetic data. Since learning a sequential model in a non-sequential setting is the primal motivation of this paper, I think it is essential to validate the algorithm on a real life data. Moreover, a performance comparison with a more conventional learning algorithm is vital for the justification of the algorithm.

-In section 4, it is indicated that as the number of data items N increases, the take off point in Figure1(a) gets closer to the true value r=0.3. I personally can not understand why the projection error is not zero when r is less than 0.3 and it is zero when r is larger. Shouldn’t it be different from zero whenever r is not equal to 0.3? A comment on this issue would be helpful.
Minor comment: On legend of figure 1, shouldn’t the logarithms be of base 10, instead of 2?

-The general motivation behind choosing spectral learning algorithms for learning latent variable models over ML based approaches is also due to their speed. There is no mention regarding the proposed algorithm on this aspect. A speed comparison with a ML based approach (possibly EM) would be beneficial. It is also possible to use matrix eigendecomposition based algorithm of Anandkumar et al. (A method of moments for mixture models and HMMs, 2012) which would be computationally cheaper than the tensor decomposition approach.
Summary: This is a well written and executed paper for a fairly interesting problem.
Author Feedback

Author rebuttal: Reviewer 1:

Thanks for pointing out recent work on spectral learning using non-sequence data. Our submission differs from that work in two aspects:

1) In our setting only non-sequence data are available, while their approach needs some sequence data as input. Although one can still apply the method by [Huang and Schneider, 2013] in the fully non-sequential setting, it is not difficult to see that the learnt model would degenerate to the stationary distribution of the HMM.

2) Our method learns the parameters of the underlying HMM, while their approach learns an observable representation of the HMM.

We will add these to the paper. We will also improve the presentation of the
finite-sample bounds to better convey its asymptotic behavior.

Reviewer 2:

Thanks for the suggestion of shrinking Section 2 and moving Appendix A.1 to the main body. We will re-organize the paper in a future version to better high-light the main ideas.

Experimental comparison with EM-based algorithms is certainly interesting, and we will definitely work on that.

Reviewer 3:
Thanks for the suggestion of presenting the search heuristics as pseudo code and shrinking Section 2. We will revise the paper accordingly.

Indeed \pi = \alpha / \alpha_0. We will make the suggested change.

Regarding the results in Section 4, the behavior of the projection distance is indeed consistent with Theorem 3, which implies that the projection distance should be (close to) zero for ALL r >= r*, the true value of r, and should be positive for r < r*. Therefore, we can determine r* by locating where the projection distance starts to increase. The legend of Figure 1(a) is of base 2 because our sample size N is in the range {1000(2^0), 1000(2^1), 1000(2^2), ..., 1000(2^10)}.